

# Revised estimate of particulate emissions from Indonesian peat fires in 2015

Kiely, Laura[1], Spracklen, Dominick V.[1], Wiedinmyer, Christine[2], Conibear, Luke[1], Reddington, Carly L.[1], Archer-Nicholls, Scott[3], Lowe, Douglas[4], Arnold, Stephen R.[1], Knote, Christoph[5], Khan, Md Firoz[6], Latif, Mohd Talib[7], Kuwata, Mikinori[8, 9, 10], Sri Hapsari Budisulistiorini[8], Syaufina, Lailan[11]

1. School of Earth and Environment, University of Leeds, Leeds, UK

2. CIRES, University of Colorado, Boulder, Colorado, USA

3. Department of Chemistry, University of Cambridge, Cambridge, UK

4. University of Manchester, Manchester, UK

5. Ludwig-Maximilians University, Munich, Germany

6. Department of Chemistry, University of Malaya, Malaysia

7. School of Environmental and Natural Resource Sciences, University Kebangsaan Malaysia, Malaysia

8. Earth Observatory of Singapore and Asian School of the Environment, Nanyang Technological University, Singapore 639798, Singapore

9. Campus for Research Excellence and Technological Enterprise (CREATE) Program, Singapore

10. Center for Southeast Asian Studies, Kyoto University, Kyoto, Japan

11. Faculty of Forestry, Bogor Agricultural University (IPB), Bogor, Indonesia

*Correspondence to*: Laura Kiely (eelk@leeds.ac.uk)

**Abstract.** Indonesia contains large areas of peatland which are being drained and cleared of natural vegetation, making them susceptible to burning. Peat fires emit considerable amounts of carbon dioxide, particulate matter (PM) and other trace gases, contributing to climate change and causing regional air pollution. However emissions from peat fires are uncertain due to uncertainties in emission factors and burn depth of peat. We used the Weather Research and Forecasting model with chemistry, and measurements of PM concentrations to constrain PM emissions from Indonesian fires during 2015, one of the largest fire seasons in recent decades. We estimate $PM_{2.5}$ (particles with diameters less than 2.5 µm) emissions from fires across Sumatra and Borneo during September to October 2015 were 7.33 Tg, a factor 3.5 greater than those in Fire Inventory from NCAR (FINNv1.5), which does not include peat burning. We estimate similar dry fuel consumption and $CO_2$ emissions to those in



the Global Fire Emissions Database (GFED4s), but a factor 1.8 greater $PM_{2.5}$ emissions, due to updated $PM_{2.5}$ emission factors for Indonesian peat. Through comparing simulated and measured PM concentrations, our work provides an independent confirmation of these updated emission factors. We estimate peat burning contributes 71% of total $PM_{2.5}$ emissions from fire in Indonesia during September-October 2015. We show that using satellite-retrieved soil moisture to modify the assumed depth of peat burn improves the simulation of PM, increasing the correlation between simulated and observed PM from 0.48 to 0.56. Overall, our work suggests that peat fires in Indonesia produce substantially greater PM emissions than estimated in current emission datasets, with implications for the predicted air quality impacts of peat burning.

## 1 Introduction

Vegetation and peatland fires across Indonesia result in habitat and biodiversity loss, large emissions of carbon and regional haze episodes. Fire events cause regional reductions in visibility and severe air pollution (Reddington et al., 2014; Gaveau et al., 2014; Kim et al., 2015; Lee et al., 2017) with associated morbidity and mortality (Marlier et al., 2012; Reddington et al., 2015; Crippa et al., 2016).

Indonesia contains 47% of total tropical peatland, the largest of any country (Page et al., 2011). Undisturbed peatlands typically have high moisture content, making them naturally resilient to fire (Wösten et al., 2008). Indonesian peatlands are experiencing deforestation and conversion to agriculture, oil palm and timber plantations (Hansen et al., 2013; Gaveau et al., 2014; Miettinen et al., 2017). During this conversion, drainage canals are installed, lowering the water table and making the peatland more susceptible to burning (Konecny et al., 2016). Fire is also used as an agricultural tool to clear vegetation (Page et al., 2002; Carlson et al., 2012). These human disturbances can make peatlands particularly prone to fire. In 2015, 53% of fires in Indonesia occurred on peatland, which made up only 12% of the land area (Miettinen et al., 2017).

Peatlands have thick organic soil layers up to 10 m deep (Hu et al., 2018). Fires on peatland can burn into these underground organic layers and smoulder for weeks after the surface fire has been extinguished (Roulston et al., 2018), resulting in substantially greater emissions compared to surface vegetation fires (Heil et al., 2007). Peat fires are estimated to contribute 3.7% of global fire carbon emissions (van der Werf et al., 2017). In Indonesia, peatland fires are the largest contributor to fire emissions in the region (Van Der Werf et al., 2010; Reddington et al., 2014). For the fires in 2015, Wooster et al. (2018) found



that 95% of the particulate matter ($PM_{2.5}$) emissions came from peat fires, and Wiggins et al. (2018) estimated that 85% of smoke plumes detected in Singapore originated from peat fires.

Whilst it is known that emissions from peatland fires are substantial, current emissions estimates have large uncertainties.
Emission estimates are typically based on remote sensed information from satellites on the area burned by the fires. Burned area may be underestimated in SE Asia due to extensive cloud cover (Ge et al., 2014).  Furthermore, estimates of burned area are limited to surface fires and may miss fires that burn underground (Kaiser et al., 2012). For peat fires, the amount of biomass consumed by the fire depends on how deep into the peat the fire burns (Hu et al., 2018). Burn depth is variable, with some fires recorded as burning to a depth of 0.85 m, resulting in carbon emissions of 31.5 kg C $m^{-2}$  (Page et al., 2002; Page and
Banks, 2007). Burn depth depends on the water content of the peat, with increased burn depth when the peat dries out (Rein et al., 2008; Huang and Rein, 2015). Konecny et al. (2016) also suggest that burn depth changes based on the frequency of fire, with reduced burn depth for repeat fires at the same location. Information on the spatial and temporal variability of burn depth is limited and current emission datasets make broad assumptions regarding these parameters. Emission factors (EFs), estimated from field or laboratory measurements, are used to convert mass of fuel consumed by the fire to the emitted mass of
gas phase and particulate pollutants (e.g. Andreae and Merlet, 2001; Akagi et al., 2011). Compared to flaming combustion, smouldering peat fires have colder combustion temperatures, and typically higher EFs for products of incomplete combustion including CO, $CH_4$, $CO_2$, HCN, $NH_3$ and PM (Hu et al., 2018). Until recently there have been few specific measurements of EFs for tropical peat fires. Roulston et al. (2018) and Wooster et al. (2018) found that assumed EFs for tropical peat fires could be underestimated by a factor of three. There are large variations in EFs for peat in Indonesia. In one study measuring emissions
from peat fires in Central Kalimantan during 7 days in 2015, $PM_{2.5}$ EFs was found to vary between 6 and 30 g $kg^{-1}$ (Jayarathne et al., 2018). EFs can also vary between years, Kuwata et al. (2018) used measurements from Indonesian peat fires to estimate EFs of $PM_{10}$ of 13±2 g $kg^{-1}$ in 2013 and 19±2 g $kg^{-1}$ in 2014.

These uncertainties cause corresponding uncertainty in estimates of emissions from peat fires, and impacts on the regional air pollution. Previous studies have scaled particulate fire emissions from global fire emission datasets, or simulated fire-derived aerosol optical depth (AOD) or PM, by a factor 1.36 – 3.00 in order to match observations (Ward et al., 2012; Johnston et al., 2012; Tosca et al., 2013; Marlier et al., 2014; Reddington et al., 2016; Koplitz et al., 2016). This suggests that particulate emissions from tropical peatland regions are underestimated in current fire emission datasets.

Severe fire events in Indonesia occur during periods of drought (van der Werf et al., 2008; Tosca et al., 2011; Gaveau et al., 2014; Field et al., 2016), resulting in strong seasonal and interannual variability. Severe droughts lower the water table, making peatlands increasingly susceptible to burning. Extensive fires and regional haze episodes across Indonesia have occurred in 1997, 2006, 2009, 2013 and 2015. During September to October 2015, dry conditions caused by a strong El Niño, resulted in



large fires across Sumatra and Kalimantan. This fire episode was the largest in Indonesia since 1997 (Huijnen et al., 2016), releasing an estimated 149±71 TgC (Jayarathne et al., 2018) to 188±67 TgC (Huijnen et al., 2016) as $CO_2$. The fires also emitted substantial amounts of $PM_{2.5}$ estimated at between 6.5±5.5 Tg (Jayarathne et al., 2018) and 9.1±3.2 Tg (Wooster et al., 2018). Particulate air pollution from these fires may have been responsible for between 6,513 and 17,270 excess deaths

through short term exposure to fire-sourced $PM_{2.5}$ (Crippa et al., 2016) and as many as 100,300 excess deaths over the longer term due to exposure to this pollution (Koplitz et al., 2016).

Given the importance of peatland fires as the main contributor to fire emissions in Indonesia, there is a high priority in reducing the large uncertainties in these emissions. In this study we aim to improve understanding of the emissions from peat fires in

Indonesia by combining fire emission datasets, a regional air quality model and extensive measurements of PM. We focus on the large fires of September to October 2015. We updated an existing fire emissions dataset to include emissions from peat fires, applying updated information on emission factors from tropical peat combustion and using satellite-retrieved information on soil moisture to control assumed depth of peat burn. We used the existing and new emissions datasets with an air quality model, and evaluate simulated PM concentrations against observations. The new emissions dataset demonstrates a substantial

improvement in simulating regional $PM_{2.5}$ concentrations.

## 2 Methodology

We used a regional atmospheric model to simulate PM concentrations during August - October 2015, with different combinations of peat and vegetation fire emissions, described below. Our study region included Borneo, Sumatra and mainland

Malaysia (Fig. 1, 95-120°E and 10°S-10°N), which is at the centre of the model domain. We used surface observations of PM and AOD to assess the performance of the model with the different fire emissions.

### 2.1 WRF-chem Model

We used the Weather Research and Forecasting model with Chemistry (WRF-chem), version 3.7.1. WRF-chem simulates gas-phase chemistry and aerosol processes fully coupled to the meteorology (Grell et al., 2005). The model was run at 30 km horizontal resolution with 33 vertical levels over 140x140 grid points centred at 110°E 0°N (90-130°E and 17°S -18°N), with Mercator projection. Simulations were run over the period of the 18th July until the 1st November. The MOZART (Model for Ozone and Related Chemical Tracers, version 4; Emmons et al., 2010) chemistry scheme was used to calculate gas-phase

chemical reactions, with aerosol dynamics and processes represented by MOSAIC (Model for Simulating Aerosol Interactions and Chemistry; Zaveri et al., 2008; Hodzic and Knote, 2014). Within MOSAIC, 4 aerosol bin sizes were used; 0.039-0.156





µm, 0.156 – 0.625 µm, 0.625 – 2.5 µm and 2.5 – 10 µm. Anthropogenic emissions were from EDGAR-HTAP2 (Janssens-Maenhout et al., 2015) for 2010, and biogenic emissions were from MEGAN (Model of Emissions of Gases and Aerosols from Nature; Guenther et al., 2006). A similar model setup has been used for studies in India (Conibear et al., 2018), the United States (Knote et al., 2014) and Indonesia (Crippa et al., 2016). The meteorology was reinitialised each month with NCEP GFS

(NCEP, 2007), with a 24 hour spin-up, and was then free-running through the month. More information on the chemistry and physics options used can be found in Table S1 in the supplement.

## 2.2 Fire emissions

We applied four different emission datasets in the WRF-chem model, all based on the Fire Inventory from NCAR (FINNv1.5).

All emission datasets included emissions from vegetation fires as treated in FINNv1.5, but with different treatment of peat combustion, as described below.

    1.   FINN (FINN)
The Fire Inventory from NCAR (FINNv1.5) combines data on active fires, biomass burned and EFs to give daily fire emissions

at 1 km resolution (Wiedinmyer et al., 2011). Vegetation burned is assigned based on the MODIS Land Cover Type and Vegetation Continuous Fields (VCF) products. Fire area burned is assumed to be 1 km² (100 ha) (scaled back by any non-vegetated area assigned by the VCF product). Fuel loading is from Hoelzemann et al. (2004) and EFs are from Akagi et al. (2011), Andreae and Merlet (2001) and McMeeking et al. (2009). FINNv1.5 includes emissions from combustion of above-ground vegetation but does not include emissions from combustion of peat.

    2.   FINN with GFED4s peat  (FINN+GFEDpeat)
In this dataset we combined vegetation emissions from FINNv1.5 with emissions from peat fires from the Global Fire Emissions Database (version 4 with small fires) (GFED4s). GFED combines burned area from Giglio et al. (2013), with assumed combustion completeness and EFs. For peat fires the depth burned is dependent on the soil moisture, with a maximum

depth of 0.5 m. GFED4s peat EFs come from Indonesian peat for $CO_2$, CO and $CH_4$, and from deforestation fires for all other species.

GFED4s data is available daily at 0.25° resolution. GFED emissions are available split by fuel type, allowing us to combine GFED4s emissions from peat fires with FINN emissions from other fuel types.

    3.   FINN with peat emissions (FINNpeat)
We created a new emissions dataset (FINNpeat), based on FINNv1.5 emissions with the addition of emissions from combustion of peat. Emissions from vegetation fires in FINNpeat are identical to those in FINN. For those fire detections



occurring on peat as identified using a peatland distribution map (WRI), additional emissions from the peat burning were calculated using Eq. (1):

$$E_s = BA \; x \; BD \; x \; \rho \; x \; EF_s \tag{1}$$

Where $E_s$ is the emissions of a species, $s$, from a fire, $BA$ is the burned area and $BD$ is the burn depth for the fire, $\rho$ is the peat density and $EF_s$ is the emissions factor for species, $s$. For each fire, the corresponding emissions are released on the day that the fire was detected, with no long-term smouldering effects.

Tansey et al. (2008) used an analysis of MODIS hotspots and MODIS burned area to estimate a 15-16 ha of burned area per hotspot. However, 60% of burned areas did not have an identified hotspot, implying an area burned per MODIS hotspot of approximately 40 ha. Over areas defined as peat we therefore assumed a burned area of 40 ha of peat burnt per hot spot, smaller than the 100 ha assumed for vegetation fires.

The mass of peat burned during peat fires was calculated from an assigned burned area, peat density and the burn depth (Table 1). We assumed a peat density of 0.11 g cm$^{-3}$ (Driessen and Rochimah, 1976; Neuzil, 1997; Shimada et al., 2001; Warren et al., 2012) and a burn depth of 37 cm for all fires detected (Page et al., 2002; Usup et al., 2004; Ballhorn et al., 2009). We assumed that all peat within the burned area and depth is combusted, as is assumed in GFED3 (van der Werf et al., 2006). This gives a fuel consumption of 40.7 kg dry matter m$^{-2}$, consistent with Leeuwen et al. (2014) who found the average fuel
consumption for Indonesian peat fire to be 31.4 kg dry matter m$^{-2}$ (from studies by Page et al., 2002; Usup et al., 2004; Ballhorn et al., 2009).

We assigned the average EFs from previous studies (Table 2) (Christian et al., 2003; Hatch et al., 2015; Stockwell et al., 2016; Wooster et al., 2018; Jayarathne et al., 2018; Nara et al., 2017; Smith et al., 2018): $CO_2$ (1670 g kg$^{-1}$), $PM_{2.5}$ (22.3 g kg$^{-1}$),
organic carbon (OC) (11.5 g kg$^{-1}$) and black carbon (BC) (0.07 g kg$^{-1}$). By comparison, GFED4s assumes similar EFs for $CO_2$ (1703 g kg$^{-1}$) and BC (0.04 g kg$^{-1}$), but substantially lower EFs for $PM_{2.5}$ (9.1 g kg$^{-1}$) and OC (6.02 g kg$^{-1}$).

The variation in measured EFs vary widely depending on the emitted pollutant, 20% for $CO_2$ (1507-1775 g kg$^{-1}$), a factor 2-3 for $PM_{2.5}$ (17.3-28.0 g kg$^{-1}$) and OC (6.02 – 16.0 g kg$^{-1}$), and an order of magnitude for BC (0.006 – 0.134 g kg$^{-1}$).  The EFs
used by Wooster et al. (2018) for $PM_{2.5}$ and $CO_2$ are  at the upper end of the ranges of EFs





**Table 1: Values for peat burn depth and peat density found in previous studies, and the average value across studies. All studies were based in Kalimantan, Indonesia.**

| | Burn depth (m) | Peat density (g/cm$^3$) |
|---|---|---|
| Page et al 2002 | 0.51 | |
| Ballhorn et al 2009 | 0.33 | |
| Centre for international co-operation in measurement of tropical peatlands (From Ballhorn et al, 2009) | 0.3 | |
| Usup et al 2004 | 0.35 | |
| Neuzil, S. G. Biodiversity and Sustainability of Tropical Peatlands | | 0.093 |
| Driessen and Rochimah 1976 | | 0.11 |
| Warren et al 2012 | | 0.127 |
| Shimada et al 2001 | | 0.112 |
| **Average** | **0.37** | **0.11** |

considered for this study. Substantial uncertainty in BC emissions has implications for the climate impacts of the aerosol, but since BC only makes a minor contribution to overall mass it has less importance for simulation of PM$_{2.5}$ .

    4.    FINN new peat with soil moisture (FINNpeatSM)

As peat dries out the burn depth increases (Usup et al., 2004; Rein et al., 2008; Wösten et al., 2008). However, FINNpeat

assumes a constant peat burn of 37 cm depth regardless of soil moisture. FINNpeatSM emissions were calculated in the same way as FINNpeat emissions, but with peat burn depth varying dependent on surface soil moisture.

Daily soil moisture from the European Space Agency (ESA CCI SMv04.4) was used to estimate burn depth in peatlands (Liu et al., 2012; Dorigo et al., 2017; Gruber et al., 2017). Frequent cloud cover leads to numerous missing values in the daily soil

moisture data at 0.25° resolution. To help account for this soil moisture was averaged to 2° resolution. In 2015, average daily soil moisture across peatlands in the study area declined from around 0.24 m$^3$ m$^{-3}$ in August, to 0.23 m$^3$ m$^{-3}$ in September to a minimum of around 0.22 m$^3$ m$^{-3}$ in October 2015, then increasing to 0.25 m$^3$ m$^{-3}$ in November  (Fig. S1). By comparing the temporal change in soil moisture over high fire regions in Sumatra and Kalimantan, we chose upper and lower limits of 0.25 m$^3$m$^{-3}$ and 0.15 m$^3$m$^{-3}$, which reflected the soil moisture in these regions before and during the dry season (Fig. S2).



**Table 2: Emission factors from previous studies, in g/kg, and an average value across all studies.**

| Method | Christian et al 2003 | Wooster et al 2018 | Stockwell et al 2016 | Stockwell et al 2015 | Hatch et al 2015 | Jayarathne et al 2018 | Smith et al 2017 | Nara et al 2017 | Average |
|---|---|---|---|---|---|---|---|---|---|
| | Lab | In situ | In situ | In situ | lab | In situ | In situ | In situ | |
| $CO_2$ | 1703 | 1775 | 1564 | 1507.23 | | | 1579 | 1663 | **1669** |
| CO | 210.3 | 279 | 291 | 224.66 | | | 251 | 205 | **243.48** |
| $CH_4$ | 20.80 | 7.9 | 9.51 | 11.69 | | | 11.00 | 7.6 | **11.17** |
| $C_2H_2$ | 0.06 | | 0.12 | | | | 0.06 | | **0.11** |
| $C_2H_4$ | 2.57 | | 0.96 | 1.09 | | | 2.30 | | **1.60** |
| $C_5H_8$ | | | 0.0528 | 1.1382 | | | | | **0.5823** |
| $CH_3OH$ | 8.23 | | 2.14 | 3.78 | | | | | **4.48** |
| HCHO | 1.40 | | 0.867 | 1.532 | | | 0.77 | | **1.220** |
| $C_2H_4O_2$ | 1.59 | | 0.108 | | | | | | **0.849** |
| $CH_3CHO$ | 3.27 | | 0.697 | 1.496 | | | | | **1.740** |
| HCOOH | 0.79 | | 0.18 | 0.53 | | | 0.25 | | **0.46** |
| $C_3H_6O$ | 1.5 | | 0.69 | 1.38 | | | | | **1.18** |
| $C_2H_6$ | | | 1.52 | | | | | | **1.52** |
| $C_3H_8$ | | | 0.989 | | | | | | **0.989** |
| $C_{10}H_{16}$ | | | 0.00167 | 0.1925 | 0.0068 | | | | **0.09984** |
| $NH_3$ | 19.92 | | 2.86 | 1.33 | | 17.3 | | | **7.09** |
| $PM_{2.5}$ | | 28.0* | 21.5 | | | | | | **22.3** |
| Black Carbon | | 0.134 | 0.00552 | | | | | | **0.0695** |
| Organic Carbon | 6.02 | | 16.0 | | | 12.4 | | | **11.5** |
| Higher Alkanes | | | 0.87 | | | | | | **0.87** |

*Contains both peat and vegetation burning





We scaled burn depth linearly from a minimum of 5 cm for a soil moisture of 0.25 $m^3m^{-3}$ to a maximum of 37 cm for soil moisture of 0.15 $m^3m^{-3}$. Under these assumptions, mean peat burn depth across peatland areas in Indonesia increased from 15.0 cm in August to 23.6 cm in September and 24.8 cm in October.

### 2.2.1 Vertical profile of fire emissions

Fires can inject emissions above the surface. By default in WRF-chem, the vertical distribution of fire emissions uses a plume rise parameterization based on a 1d cloud model (Freitas et al., 2007). However recent work suggests that tropical fires mostly inject emissions into the BL and the WRF-chem scheme may overestimate fire injection heights. Tosca et al. (2011) found that the average plume height for fires in Sumatra and Borneo was 729 m, with 96% of plumes confined to within 500 m of the boundary layer. Martin et al. (2018) found that 90% of fire emissions in South Asia in September to November were injected below 1500 m. Archer-Nicholls et al. (2015) found that the WRF-chem plume rise parameterisation overestimated the injection height for fires in South America. For this reason we chose not to use the plume-injection option and instead tested two alternate approaches to control the vertical profile of fire emissions:

- All of the emissions were added to the surface model layer (surface injection),
- Half of the emissions were added to the surface model layer and 50% of the emissions were spread evenly to model layers throughout the boundary layer (boundary layer injection)

### 2.3 Particulate measurements

Measurements of particulate matter with diameter less than 2.5 µm ($PM_{2.5}$), less than 10 µm ($PM_{10}$), and less than 1 µm ($PM_1$), and measurements of AOD (Table 3) were used to evaluate the model. Figure 1 shows the locations of measurements. Hourly measurements of $PM_{2.5}$ concentrations are available from the National Environment Agency of Singapore for five sites in Singapore during October 2015. We averaged concentrations across the five sites to produce mean $PM_{2.5}$ concentrations for Singapore. From Singapore, there are also measurements of non-refractory, composition resolved sub- micron PM from an Aerosol Chemical Speciation Monitor (ACSM) (Budisulistiorini et al., 2018). We summed the chemically-resolved masses to give $PM_1$. Betha et al. found that for fire-induced haze in Singapore in 2013, 96-99% of the $PM_{2.5}$ was $PM_1$.

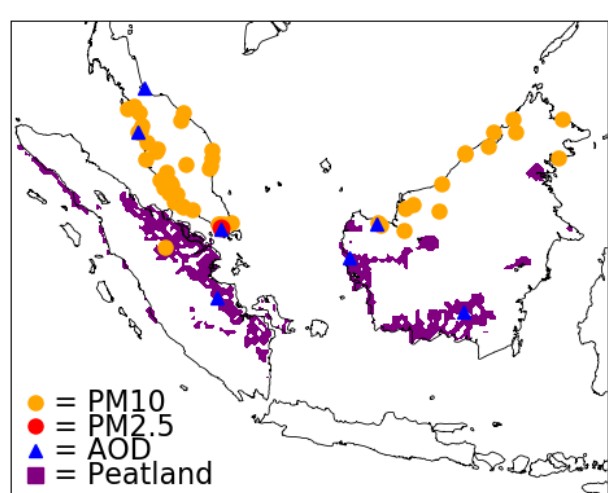

**Figure 1: The study area showing the locations of $PM_{10}$ measurements in yellow circles, $PM_{2.5}$ in red circles and AOD in blue triangles. Peatland is shown in purple.**




**Table 3: Observational data for 2015.**

| Data | Location | Time Period | Frequency of observations | Method | Reference / Source |
|------|----------|-------------|---------------------------|--------|---------------------|
| $PM_{2.5}$ | 5 Sites in Singapore: N: 1.41°N, 103.80°E E:1.33°N, 103.69°E C:1.34°N, 103.82°E W:1.33°N, 103.93°E S: 1.27°N, 103.83°E | 28 Sep 2015– 15th Nov 2015 | 1 hr | Thermo Scientific™ 5030 SHARP Monitor | National Environment Agency for Singapore |
| Composition resolved non-refractory $PM_1$ | National Technological University in Singapore 1.35°N, 103.68°E | 10 Oct 2015 – 31 Oct 2015 | 2-3 mins | Aerosol Chemical Speciation Monitor (ACSM) | Budisulistiorini et al. (2018) |
| $PM_{10}$ | Pekanbaru, Indonesia 0.52°N, 101.43°E | 1 Jan 2010 – 31 Dec 2015 | 30 mins | Measured using a Met One BAM 1020, Real-Time Portable Beta Attenuation Mass Monitor (BAM-1020) | |
| $PM_{10}$ | 52 locations across Malaysia | Aug-Nov 2015 | 1 hr | Measured using a Met One BAM 1020, Real-Time Portable Beta Attenuation Mass Monitor (BAM-1020) | Mead et al., 2018 |
| AOD | 8 AERONET sites | Aug - Oct 2015 | 24 hr average | Ground based remote sensing Sun photometer instrument measures intensity of solar radiation at 500nm wavelength, from which AOD is derived | AERONET version 2 |



In October 2015, measured $PM_1$ agreed to within 20% of the mean $PM_{2.5}$ concentration from the NEA. Ground-based AOD measurements were available from 7 Aerosol Robotic Network (AERONET) sites for August to October. Measurements of hourly $PM_{10}$ were available from 52 locations across Malaysia (Mead et al., 2018) and one location in Indonesia during August to October 2015. We compared daily mean observations at each site with simulated PM and AOD in section 3.2. The fractional

bias (defined in supplementary information) and correlation coefficients were used to evaluate the simulations. We did not use AOD data from MODIS retrievals, which significantly underestimated AOD over the region during this period, due to excluding smoke plumes that were mistook for clouds (Shi et al., 2019).

## 3 Results

### 3.1 Fire emissions

Table 4 shows total dry matter consumption, $PM_{2.5}$ and $CO_2$ emissions from fires across Sumatra and Borneo in September and October 2015. The dry fuel consumption is lowest for FINN (230 Tg), which does not include peat fires. Dry matter consumption is similar for GFED, FINN+GFEDpeat and FINNpeatSM (455 Tg, 514 Tg, 465 Tg respectively), and is highest for FINNpeat (612 Tg). This is likely due to the peat burn depth being greatest for FINNpeat. Wooster et al. (2018) estimated

358±107 Tg of dry matter consumption for Kalimantan and Sumatra in September and October, in agreement with GFED and FINNpeatSM. FINN estimates a smaller dry matter consumption compared to Wooster et al. (2018), whereas FINNpeat estimates greater dry matter consumption.

Total September and October 2015 emissions of $CO_2$ follow a similar pattern to dry matter consumption, with similar values

for GFED, FINN+GFEDpeat and FINNpeatSM (773 Tg, 822 Tg and 781 Tg), largest emissions for FINNpeat (1014 Tg) and smallest emissions for FINN (353 Tg). $CO_2$ EFs, are similar for GFED and FINN peat (1703 g kg$^{-1}$ and 1669 g kg$^{-1}$), explaining the similarity between dry matter consumption and $CO_2$ emissions for these datasets. The total $CO_2$ emissions for September to October estimated by Wooster et al., (2018) was 692±213 Tg, matching GFED, FINN+GFEDpeat and FINNpeatSM. Jayarathne et al., (2018) estimated 547±259 Tg of $CO_2$ were emitted over South Sumatra and Kalimantan,

suggesting that the total $CO_2$ emissions from FINN (353 Tg for Borneo and Sumatra) are too small, due to lack of peat fires in this dataset.

Total emissions of $PM_{2.5}$ vary across simulations due to differences in assumed $PM_{2.5}$ EFs. FINN has the smallest total $PM_{2.5}$ emissions for September to October (2.09 Tg; Table 4). GFED and FINN+GFEDpeat have similar total $PM_{2.5}$ emissions (4.14

30 Tg and 4.60 Tg), smaller than that for FINNpeatSM (7.33 Tg) despite these datasets having similar dry matter consumption and $CO_2$ emissions. This is due to the difference in the assumed EFs for $PM_{2.5}$ from peat fires, with 9.1 g kg$^{-1}$ used in GFED,





**Table4: Total dry matter fuel consumption, PM$_{2.5}$ and CO$_2$ fire emissions for September and October 2015. Totals are shown for the area shown in Fig. 1.The percentage contribution from peat fires is indicated.**

|  | FINN | GFED | FINN+GFEDpeat | FINNpeat | FINNpeatSM |
|---|---|---|---|---|---|
| Peat fires included | No | Yes | Yes | Yes | Yes |
| Dry matter fuel consumption (Tg) | 230 | 455 | 514 | 612 | 465 |
| CO$_2$ emissions (Tg) | 353 | 773 | 822 | 1014 | 781 |
| Contribution from peat fires | 0% | 63% | 57% | 65% | 55% |
| PM$_{2.5}$ emissions (Tg) | 2.09 | 4.14 | 4.60 | 10.60 | 7.33 |
| Contribution from peat fires | 0% | 62% | 55% | 80% | 71% |

and 22.26 g kg$^{-1}$ used in FINNpeatSM. Wooster et al., (2018), assumed a PM$_{2.5}$ EF of 28±6 g kg$^{-1}$ and estimated that 9.1±3.2 Tg of PM$_{2.5}$ was emitted over the whole of Sumatra and Kalimantan for September and October 2015, similar to that found in FINNpeat (10.60 Tg) and FINNpeatSM (7.33 Tg). In contrast, FINN and FINN+GFED, which use the lower EF, produce smaller PM$_{2.5}$ emissions by a factor of 2 and 4 respectively. Jayarathne et al. (2018) found that, for a smaller area of Sumatra

and Kalimantan, the total PM$_{2.5}$ emission was 6±5.5 Tg, a range which covers the total PM$_{2.5}$ from all of the datasets.

Table 4 also shows the fraction of CO$_2$ and PM$_{2.5}$ emissions that are estimated to come from peat fires. Across the datasets that include peat burning, peat fires contribute 51-62% of dry matter consumption, 55-65% of CO$_2$ emissions and 55-80% of PM$_{2.5}$ emissions. The emission datasets with updated PM$_{2.5}$ emission factors result in a greater contribution from peat burning (71%-

80%) compared to emission datasets with the older EFs (55%-62%). Wooster et al. (2018) found that peat fires contributed 85% of the dry matter fuel consumption, and 95% of the PM$_{2.5}$ emissions in September and October 2015, even greater than our estimates with updated EFs.

Figure 2 shows the spatial variations of the total PM$_{2.5}$ emissions during September and October 2015. In all datasets, greatest

emissions occur in southern Kalimantan and central and southern Sumatra, matching the locations of peatlands (Fig. 1). For the FINNpeatSM emissions, Sumatra contributes 42% of the total PM$_{2.5}$ emissions, for FINNpeat, FINN+GFEDpeat and FINN, the contribution is 39%, 40% and 32% respectively. Wooster et al. (2018) found that 33% of the total PM$_{2.5}$ emissions



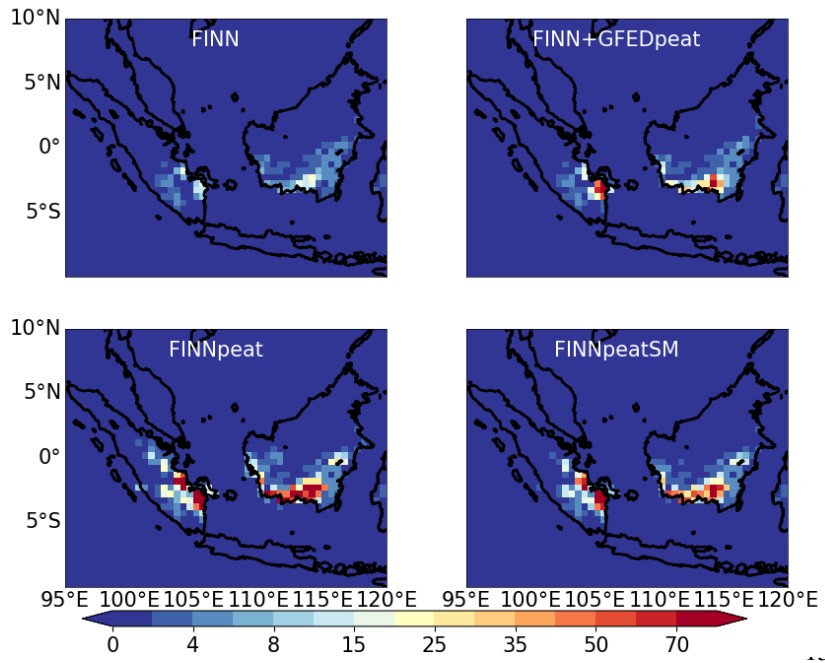

**Figure 2: Total PM$_{2.5}$ fire emissions during September-October 2015 (g /m2).**

came from Sumatra, while Koplitz et al. (2016) found that that 47% of OC and BC emitted in June to October 2015 came from Sumatra. Our estimates exclude fire emissions from eastern Indonesia. Nechita-banda et al. (2018) estimated that fires in eastern Indonesia and Papua New Guinea contributed around 15-20% of total CO emissions from fires across the region, highlighting the need future work to quantify PM emissions in this region.

FINN and GFED underestimate total emitted PM$_{2.5}$ and FINN underestimates total emitted CO$_2$ compared to the emissions found by Wooster et al. (2018) and Jayarathne et al. (2018), suggesting that peat fires are important contributors to these emissions. FINNpeatSM is the only emissions dataset that matches these previous studies for both PM$_{2.5}$ and CO$_2$.

Figure 3 shows daily total PM$_{2.5}$ emissions from the different datasets over the study area.    Temporally, the datasets follow a similar pattern, with 80-90% of the total PM$_{2.5}$ emissions for 2015 occurring in August-October. For all the emissions datasets the majority of emissions are in September, followed by October and then August. GFED has the largest difference between September and October emissions (58% in Sep and 17% in Oct), followed by FINN+GFEDpeat (47% and 24%), FINNpeat (36% and 30%), and finally FINN (33% and 29%) and FINNpeatSM (36% and 32%) which have the smallest differences between the two months. The reduced ratio of the fraction of emissions in September compared to October for FINNpeatSM is due to greater soil moisture in September resulting in a reduced peat burn depth.

### 3.2 Comparison of model and observational data

We evaluated the WRF-chem simulations with the different emissions datasets and injection options against measured PM$_{10}$, PM$_{2.5}$ and PM$_1$ concentrations. Figures 4 and 5 shows the comparison of simulated and observed PM concentrations. Comparisons of PM$_{2.5}$ and PM$_1$ measurements only, which were restricted to Singapore, are shown in Fig. S3.

PM concentrations are underestimated by the model with FINN emissions, with a fractional bias (FB) of -0.67 with surface injection and -0.77 with boundary layer injection of emissions, with an average across both simulations of -0.72. The model



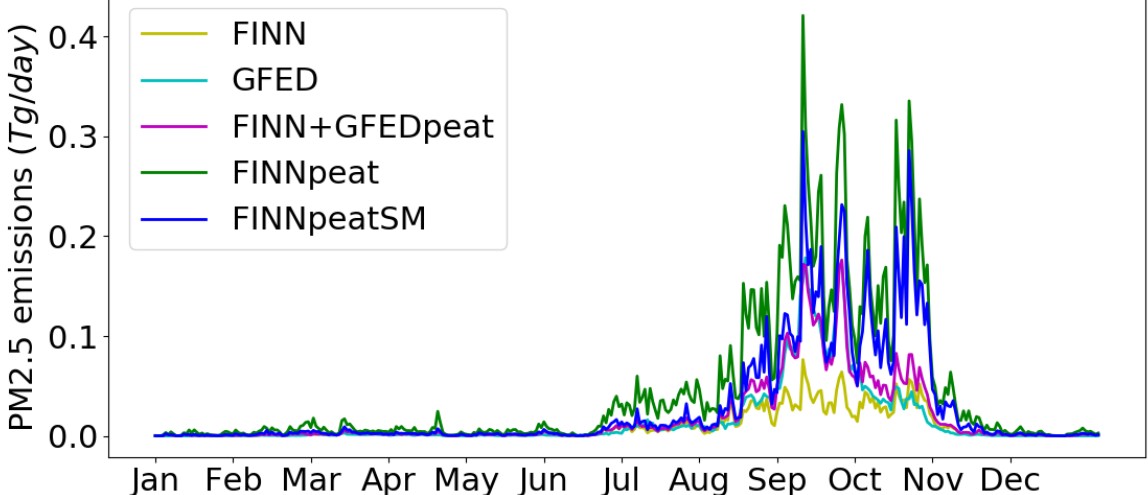

**Figure 3: Total daily PM₂.₅ emissions from fires during 2015. Total shown for the area in Figure 1.**

with FINN+GFEDpeat emissions also underestimates PM concentrations (average FB= -0.35) whilst the model with FINNpeat emissions overestimates PM concentrations (average FB=0.2). The model with FINNpeatSM emissions has the smallest bias (average FB=-0.11, suggesting mean emissions from this dataset are closest to reality.

The temporal pattern of measured PM is generally matched by the simulations, as shown in Fig. 4. However, for many sites, the greatest PM concentrations were measured in October, whereas the model simulates greatest PM concentrations in September. This results in the model underestimating PM concentrations the most in October, with a smaller underestimate, or an overestimate in September (Fig. 5b).

Using a burn depth dependant on soil moisture alters the temporal pattern of simulated emissions, reducing the overestimation in September compared to October. When burn depth is constant, as in FINNpeat, 37% of regional PM₂.₅ emissions for 2015 occur in September and 30% in October. In FINNpeatSM, where we assume a linear relationship between soil moisture and burn depth, the percentage of annual PM₂.₅ emissions in September is 39% and 36% in October. A non-linear relationship between soil moisture and burn depth, would result in shallower burn depth in September and deeper burn depth in October,

decreasing emissions in September and increasing emissions in October, which might further improve simulated PM concentrations. There is little information available on the measured relationship between soil moisture and burn depth.

The overestimation of modelled PM concentrations in September may also be due to our assumption that all the emissions from a fire are emitted on the day the fire was detected. In reality, peat fires can smoulder for weeks, and the emissions should

be released over a longer time period. This could also reduce the simulated PM concentration in September and increase them




**Figure 4: Daily observed and modelled (a) PM$_{2.5}$ in Singapore, and (b) PM$_{10}$ in Pekanbaru, for WRF-chem runs with different fire emissions datasets and the surface injection option. a) shows observations of PM$_{2.5}$ (solid) and PM$_1$ (dashed).**

in October. The overestimation in September could also be due to an issue with fire detection. Syaufina and Sitanggang (2018) found that only hotspots which last for 3 days indicate fires, something which is not considered when calculating the emissions. However, despite our simplified assumptions the model captures individual peaks in measured PM reasonably well (Fig. 4).




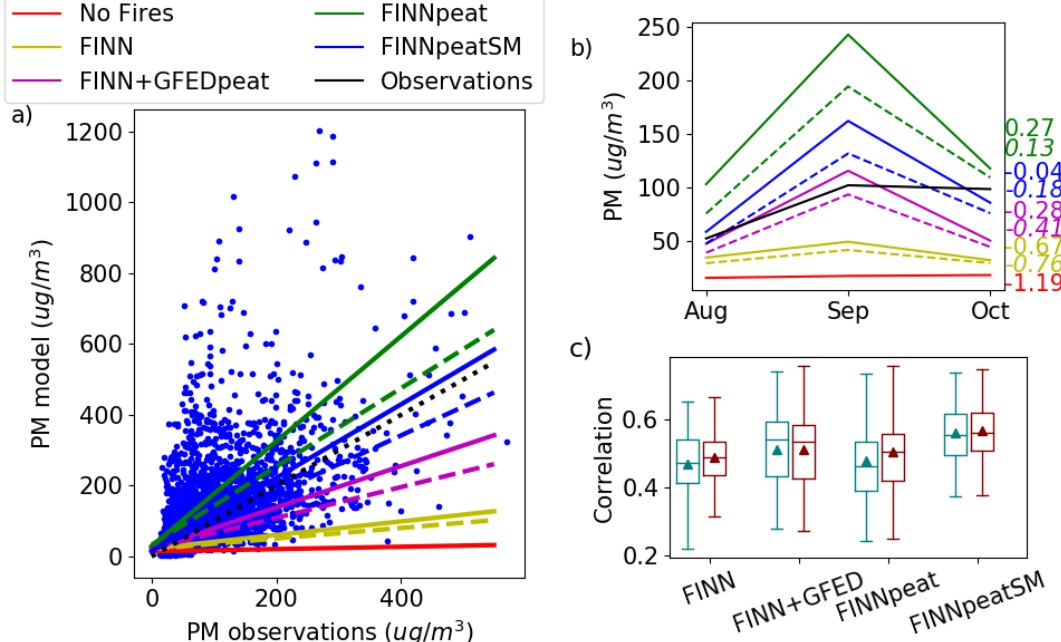

**Figure 5: Comparison of simulated and observed PM concentrations during August to October 2015. Observations of PM$_{10}$, PM$_{2.5}$ and PM$_1$ from 55 sites in Indonesia, Singapore, and Malaysia. (a) Simulated and observed daily mean PM concentrations for FINNpeatSM emissions and surface injection (blue dots). Lines show the linear fit for the model with different emissions, solid lines are when emissions are emitted at the surface, dashed lines when emissions are injected into the boundary layer. The 1:1 line is shown in black dots. (b) The average monthly simulated and observed PM concentrations. The fractional bias for August to October is shown to the right of each line. (c) The correlation coefficient (r) for comparisons of daily mean simulated and observed PM concentrations for all 55 sites. For each simulation the box plots show the median (middle line of box), upper and lower quartiles (top and bottom of box), and the range of correlations (whiskers extend to min and max) across all sites are shown by the box plots, and the mean correlations are shown as triangles. Simulations with the surface injection are in light blue, and simulations with the boundary layer injection are in red.**

Putriningrum et al. (2017) found that WRF-chem with FINNv1 or GFEDv4 emissions underestimated PM concentrations across Indonesia during October 2015, with GFEDv4 resulting in a better match compared to FINNv1. Putriningrum et al. (2017), suggested that emissions were underestimated due to haze from fires blocking the detection of burned area. In contrast,

5    our work suggests that PM emissions in GFED4 are underestimated because EFs for peat combustion are too small.

Figure 5 also shows the correlation coefficients between model and measured PM concentrations across all the observation sites. The FINN simulation has the lowest average correlation across all sites (r=0.47 and 0.49 with surface and boundary layer injection respectively), followed by FINNpeat (r = 0.48 and 0.51) and FINN+GFEDpeat (r = 0.51 for both injections).

10   FINNpeatSM has the highest average correlation across all the sites (r = 0.56 to 0.57). Both FINN+GFEDpeat and FINNpeatSM, assume variable peat burn depth depending on soil moisture. This comparison therefore suggests that varying depth of peat burn based on surface soil moisture, as in FINNpeatSM and FINN+GFEDpeat, results in improved estimate of





emissions. The height at which emissions are injected has little impact on the correlation, so there is limited evidence from this comparison to support either option.

Comparison with $PM_{2.5}$ concentrations measured in Singapore during October 2015 further supports the above analysis. WRF-chem underestimates $PM_{2.5}$ concentrations in Singapore both with FINN emissions (FB=-0.6 for surface emissions and -0.69 for boundary layer emissions) and FINN+GFED emissions (FB = -0.28 for both injections). With FINNpeat emissions the model overestimates $PM_{2.5}$ concentrations (FB = 0.45 to 0.53) and the best agreement with observations is with FINNpeatSM emissions (FB = 0.06 to 0.16).

Chemically-resolved $PM_1$ measurements from Singapore are available for $10^{th}$ to the 31st October 2015. Organic aerosols (OA) contributed 79% of the observed $PM_1$ between $10^{th}$ and $31^{st}$ October (Budisulistiorini et al., 2018). The FINN simulation underestimates the contribution of OA to $PM_1$ with 64% with BL injection (69% with surface injection). For the simulations with peat emissions, the model is improved with the contribution of OA to PM1 varying (Fig. S4). With FINNpeatSM, 78% of $PM_1$ is OA with the boundary layer injection (82%, surface injection). For the simulations with FINNpeat it is 80% (84%), for FINN+GFED 78% (79%).

Figure 6 shows comparison of simulated and measured AOD. The comparisons are consistent with that seen for PM. The model with FINN emissions underestimates AOD (FB = -0.56 for surface emissions and -0.73 for boundary layer), as does the model with FINN+GFED emissions (FB = -0.09 and -0.29). FINNpeat overestimates for both injection options (FB = 0.54 and 0.35), and FINNpeatSM gives the lowest FB of -0.003 with boundary layer injection (0.19 with surface injection). The correlation coefficients between simulated and measured AOD are highest for simulations with FINNpeatSM (r = 0.64 with surface and 0.65 with BL injection) followed by FINN+GFEDpeat (r = 0.58 and 0.59), FINNpeat (r = 0.57 for both injections), and FINN (r = 0.53 and 0.52). Previous work has found that models tend to better simulate $PM_{2.5}$ compared to AOD in regions influenced by fire emissions (Aouizerats et al., 2015; Crippa et al., 2016).

### 3.3 $PM_{2.5}$ concentrations and AOD

Figure 7 shows simulated surface $PM_{2.5}$ concentrations due to fires during September to October, (Fig. S5 shows results for the boundary layer injection). Simulated $PM_{2.5}$ concentrations from fires are greatest over Sumatra and southern Kalimantan, with simulated September-October mean concentrations exceeding 1800 µg m$^{-3}$ with FINNpeatSM emissions. Enhanced regional $PM_{2.5}$ concentrations are simulated to the north east of the fires across peninsular Malaysia (50-150 µg m$^{-3}$), caused by regional transport of pollution. Simulated surface $PM_{2.5}$ concentrations from fires during September and October over Sumatra and Borneo are greatest with FINNpeat emissions (267 µg m$^{-3}$), followed by FINNpeatSM (183 µg m$^{-3}$), FINN+GFEDpeat (98 µg m$^{-3}$) and FINN (45 µg m$^{-3}$), matching the $PM_{2.5}$ emissions from the different datasets (Table 4).




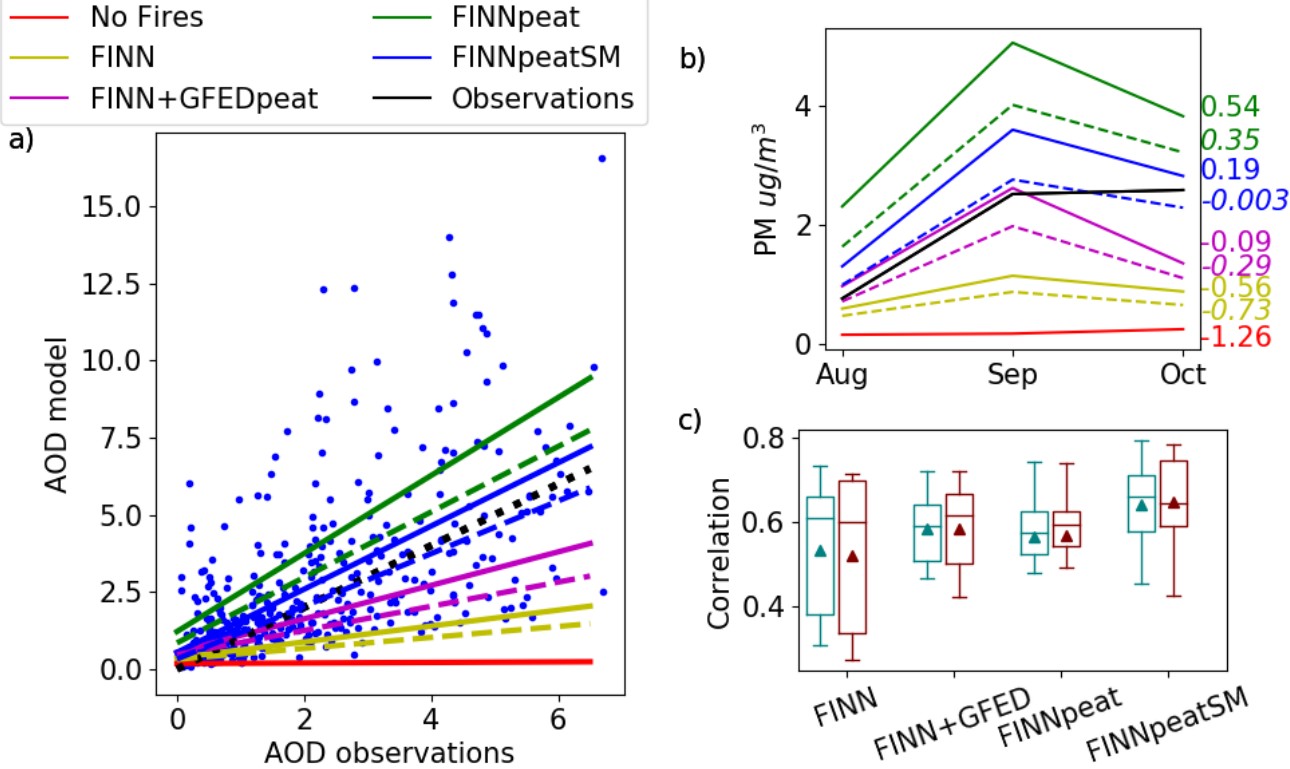

**Figure 6: Comparison of simulated and observed AOD during August to October 2015, from 8 AERONET sites in Indonesia, Singapore, and Malaysia. Observed AOD is at 500 nm and simulated AOD is at 550 nm. (a), (b) and (c) show the same as in Fig. 5, for AOD.**

Peat combustion contributes a substantial fraction of simulated $PM_{2.5}$ concentrations from fires, ranging from 55% in the model with FINN+GFEDpeat emissions, 76% with FINNpeatSM emissions to 83% with FINNpeat emissions. Figure 8 shows the fraction of the simulated surface $PM_{2.5}$ concentration from peat fires for September to October 2015 using the FINNpeatSM emissions. The majority of simulated $PM_{2.5}$ concentrations across the study area are due to emissions from peat fires. Across Sumatra and Borneo, 96% of surface $PM_{2.5}$ concentrations are from fires with 73% from peat combustion. Peat fires therefore account for 76% of the fire contribution to $PM_{2.5}$. This is slightly larger than the contribution of peat fires to primary $PM_{2.5}$ emissions (71% in FINNpeatSM), due to atmospheric production of secondary organic aerosol from fire-emitted precursors. Reddington et al. (2014) used a combination of models to demonstrate that regional fire-derived PM concentrations during haze episodes are dominated by emissions from peatland regions.

Inclusion of emissions from peat fires gives largest fire emissions in Sumatra and western Kalimantan, where previously emissions were substantially lower (Fig. 2). This leads to higher $PM_{2.5}$ concentrations across Singapore (Fig. 5), which has a





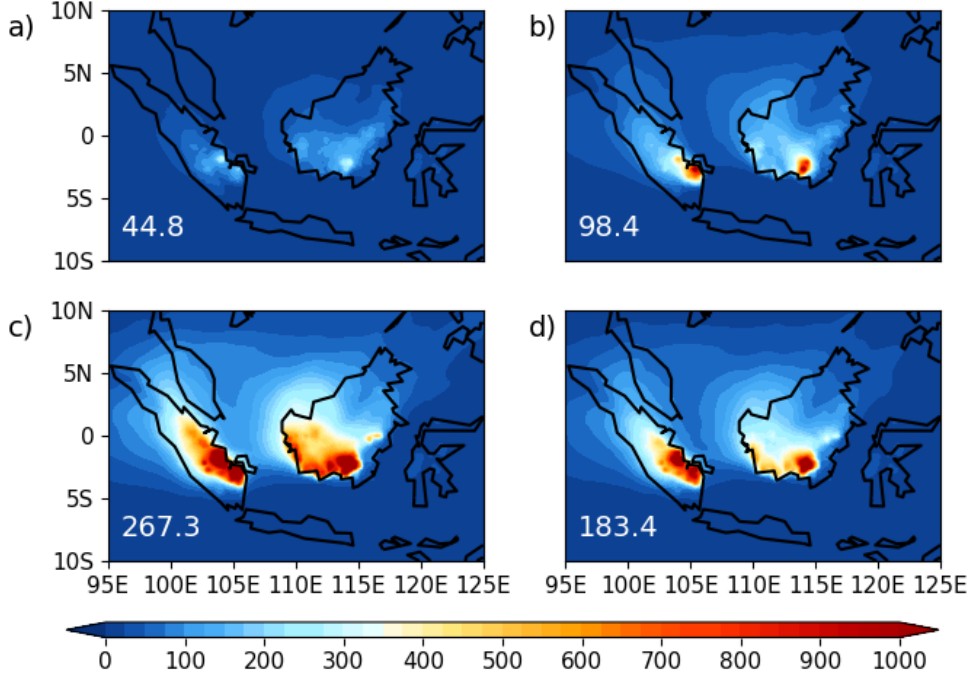

**Figure 7: Mean simulated surface PM$_{2.5}$ concentration (µg/m3) from fires for September to October 2015 with the surface injection and (a) FINN emissions, (b) FINN+GFEDpeat, (c) FINNpeat, (d) FINNpeatSM. The surface PM$_{2.5}$ concentration from fires, averaged over Sumatra, is indicated on each panel.**

large impact on the population exposure to the PM$_{2.5}$. Lee et al. (2017) found that the 2015 fires in Sumatra accounted for 50% of fire-derived PM$_{2.5}$ in Kuala Lumpur and 41% in Singapore. Reddington et al. (2014) and Kim et al. (2015) found that for the 2006 fires Sumatran fires were responsible for the worst air quality across Equatorial Asia.

Injecting all fire emissions at the surface increases the average simulated surface PM$_{2.5}$ concentration by a factor of 1.34 to 1.36 compared to injecting 50% at the surface and 50% through the boundary layer. However, this factor varies spatially (Fig. 9). Close to the fire locations, the surface injection option results in an increase in PM$_{2.5}$ concentrations by up to a factor of 2. Further away from the fires, however, the injection option has less impact on

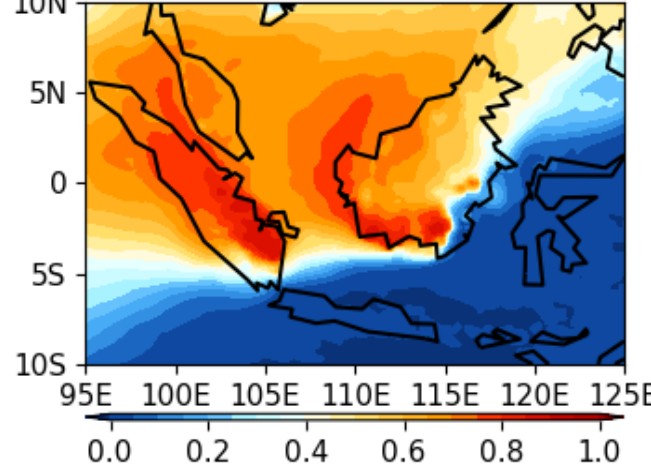

**Figure 8: Fraction of simulated PM$_{2.5}$ concentrations originating from peat fire emissions. Simulations use the new FINNpeatSM fire emissions with surface injection.**



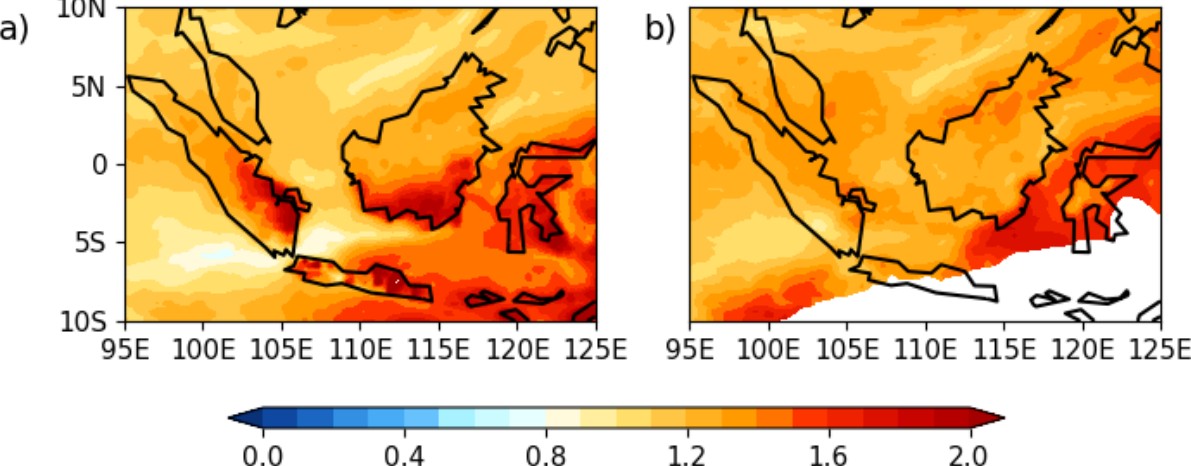

**Figure 9: Ratio of simulated (a) surface PM$_{2.5}$ concentration and (b) AOD at 550 nm from fires for September to October, when using surface injection option compared to boundary layer injection option. Results are shown for the model with FINNpeatSM emissions. Zero values of average PM$_{2.5}$ and AOD have been removed.**

simulated PM$_{2.5}$ concentrations. Despite these differences in simulated PM, the available measurements of PM do not allow us to better constrain the vertical profile of fire emissions (Section 3.2).

Simulated AOD in Sumatra and Borneo during September to October (Fig. S6) follows a similar pattern to simulated PM$_{2.5}$, with the highest value for the model with FINNpeat (1.42) followed by FINNpeatSM (1.06), FINN+GFEDpeat (0.62) and FINN (0.31). Injecting fire emissions at the surface also results in greater simulated AOD compared to when emissions are spread through the boundary layer. The average AOD across Borneo and Sumatra increases by a factor of 1.32 for surface injected emissions compared to the boundary layer, which matches the difference seen in average PM$_{2.5}$ concentrations. However, spatially the difference between the injection options is different from that seen for PM$_{2.5}$ (Fig. 8). Rather than seeing a larger increase around the fires in Sumatra and Kalimantan, the factor difference between the two injection options remains between 1.1 and 1.5 across the area effected by fires. Majdi et al. (2018) also found that the sensitivity of simulated PM$_{2.5}$ to injection method (up to 50%), was greater than the sensitivity of AOD (up to 20%), which matches the differences seen here close to fires.

## 4 Conclusions

Vegetation and peat fires in Indonesia emit substantial amounts of trace gases and aerosol resulting in serious air pollution episodes. The magnitude of emissions from these fires is very uncertain, particularly for peat fires which are more difficult to detect using Earth observation methods. New measurements of tropical peat combustion have led to an upward revision of



particulate emission factors, leading to a suggestion that fire emission datasets may underestimate particulate emissions from peat fires. Here we used the WRF-chem model along with extensive observations of PM to make a revised estimate of PM emissions from Indonesian fires during September – October 2015.

5    Current fire emission datasets either do not include peat fires, (FINNv1.5), or do not use updated peat emission factors (GFEDv4s). The WRF-chem model underestimated PM concentrations measured in Indonesia and Malaysia during August to October 2015, both with FINNv1.5 emissions (fractional bias = -0.7), and with a combination of FINN vegetation emissions and GFED4s peat emissions (fractional bias = -0.35). We created a new emissions datasets for Indonesia using updated emission factors for peat combustion and with variable assumptions relating the depth of peat burn to soil moisture (FINNpeatSM). Our best emissions estimate, FINNpeatSM, leads to an improved simulation of PM concentrations (fractional bias = -0.11). Estimated $PM_{2.5}$ emissions from fires across Sumatra and Borneo during September to October 2015 are 7.33 Tg (with FINNpeatSM), a factor 1.8 greater than in GFED4 (4.14 Tg) and a factor 3.5 greater than FINN1.5 (2.1 Tg). Since updated $CO_2$ EFs for peat fires are similar to previous measurements, our estimated $CO_2$ emissions are consistent with GFED4s.

We find that emissions from peat combustion make up a substantial fraction of total fire emissions from the region. We estimate that peat combustion contributes 55% of total $CO_2$ emissions and 71% of primary $PM_{2.5}$ emissions during September to October 2015. Peat combustion contributes 76% of fire-derived surface $PM_{2.5}$ concentrations over Sumatra and Borneo during this period. This highlights the importance of peat fires and the need for better estimates of emissions from peat combustion.

The depth of peat burn is a crucial factor controlling emissions from peat fires, but it is poorly constrained. We found that using satellite remote sensed soil moisture to control the assumed depth of peat burn improved the simulation of PM, with the correlation between simulated and measured PM increasing from 0.48 with fixed peat burn depth to 0.56 with soil moisture control. Work is now needed to examine whether this is consistent for years other than 2015.

Our work suggests that existing emission datasets (GFED4 and FINN1.5) underestimate particulate emissions from Indonesian fires, due to an underestimation of particulate emissions from peat combustion. Including updated emission factors from tropical peat combustion, results in substantially increased PM emissions from Indonesian fires. Measurements of emission factors from tropical peat combustion are still very limited, and additional measurements are required. Our comparison of
30   simulated and measured PM concentrations across the region provides an additional and independent confirmation of updated emission factors from peat combustion. Our work suggests that previous studies may have underestimated the impact of Indonesian fires on particulate air quality. We estimate that vegetation and peat fires increased PM2.5 concentrations over Sumatra and Borneo during September and October 2015 by an average of 127 $\mu$g m$^{-3}$. Future work needs to explore the impact of these fires on public health.



*Code/Data availability.* Code and data used in this study are available from authors upon request.

*Author Contributions.* Conceptualization by LK and DVS, with support from SRA and LS. LK carried out modelling work and analysis, with help from DVS and extra supervision from SRA. CK provided WRFotron modelling scripts and SAN and DL provided model code and technical support with modelling. LC and CLR provided technical support with modelling. LK and CW provided fire emissions, and MK, SHB, MFK and MTL all provided observations. LK prepared the manuscript with help from all authors.

*Competing interests.* The authors declare that they have no conflict of interest.

**Acknowledgements:**

LK was funded by a studentship from the NERC SPHERES Doctoral Training Partnership (NE/L002574/1) and by the United Bank of Carbon (UBoC). This work was supported by an Institutional Links grant, ID 332397925, under the Newton-Indonesia partnership. The grant is funded by the UK Department of Business, Energy and Industrial Strategy (BEIS) and delivered by the British Council. We acknowledge the use of the WRFotron scripts developed by Christoph Knote to automatize WRF-chem runs with re-initialized meteorology. We would like to acknowledge S.C. Liew and S.V.Cortijo at National University Singapore who run the Singapore AERONET site, Mr. Syahrial Sumin, who provided us with PM10 data from Pekanbaru, and the National Environment Agency of Singapore, for collecting and providing $PM_{2.5}$ data (available at http://www.nea.gov.sg/anti-pollution-radiation-protection/air-pollution-control/psi/historical-psi-readings).

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
