# Peer review of "Table S1: Chemistry and Physics options used in WRF-chem"

_Atmospheric Chemistry and Physics, 2019_

## Short Comment (SC1) · 29 Apr 2019

A paper published just last week in JGR appears to be relevant to your study.

Eck, T. F., Holben, B. N., Giles, D. M., Slutsker, I., Sinyuk, A., Schafer, J. S., et al. (2019). AERONET remotely sensed measurements and retrievals of biomass burning aerosol optical properties during the 2015 Indonesian burning season. Journal of Geophysical Research: Atmospheres, 124. https://doi.org/10.1029/2018JD030182

https://agupubs.onlinelibrary.wiley.com/doi/10.1029/2018JD030182

In particular, very high mid-visible AOD levels ($\sim$11 to 13 at 550 nm) were estimated from extrapolation of near-infrared measurements. These values are very similar to those of some of your model estimates as shown in your ACPD Figure 6.

[Figure]

Additionally the estimated percentage contribution of peat to AOD in Palangkaraya, Indonesia was found to be ~80% to 85% from AERONET aerosol absorption retrievals in our study, very similar to the values you show for this region in your ACPD paper Figure 8 for PM2.5.

---

## Referee Comment (RC1) · Robert Yokelson (Referee) · 6 Jun 2019

Bob Yokelson

In this work new peat fire emission factor (EF) measurements from the literature and a new approach for model estimates of peat consumption developed by the authors are incorporated into the authors WRF-chem regional model for insular SE Asia and better agreement with ambient PM monitoring and AOD are observed. The impact of injection altitude is also explored. The new peat consumption approach may be developed further, but has already reduced bias and increased correlation. A few improvements could be made to the manuscript. Mainly, a little more could be added about the impact of secondary processes and efforts made to avoid recycling some confusion on literature EFs. My suggestions are further clarified in my detailed comments provided in order of appearance in the paper in "page number, line number:" format if not otherwise specified. If, after commenting at one "P, L", I saw the issue partially addressed later in the paper, I did not delete my first comment in most cases. This is because it may be worth addressing the issue earlier in the paper?

Title: very minor point "Revised" kind of implies they are changing a previous estimate they made?

1, 22: Two minor points. 1) There is a lot of peat in Malaysian Borneo. 2) The drainage canals may have already largely been built by the early 1980s? There's a debate now about blocking them to re-flood the peatlands.

1, 25: "fuel consumption" may be better/safer than "burn depth" since the area burned is hard to measure due to small size, smoldering, high regional cloud cover, etc. (Reid et al., 2013)

Reid, J. S., Hyer, E. J., Johnson, R., Holben, B. N., Yokelson, R. J., Zhang, J., Campbell, J. R., Christopher, S. A., Di Girolamo , L., Giglio, L., Holz, R. E., Kearney, C., Miettinen, J., Reid, E. A., Turk, F. J., Wang, J., Xian, P., Zhao, G., Balasubramanian, R., Chew, B. N., Janai, S., Lagrosas, N., Lestari, P., Lin, N.-H., Mahmud, M., Nguyen, A. X., Norris, B., Oahn, N. T. K., Oo, M., Salinas, S. V., Welton, E. J., Liew, S. C.: Observing and understanding the Southeast Asian aerosol system by remote sensing: An initial review and analysis for the Seven Southeast Asian Studies (7SEAS) program, Atmos. Res., 122, 403-468, doi:10.1016/j.atmosres.2012.06.005, 2013.

2. 1: GFED4s, the "s" indicates the product with small fires added?

2, 2-3: Technically, to "confirm" an EF with a model you'd have to compare the model to the measurements at the point of emission or somehow know that the model correctly treated all secondary processes, so it's probably better to claim "consistency" or "support" for the new EF. Also the paper should mention the relative change in PM due to post emission processes predicted by the model at some point.

2, 19: Is this statistic still current now that large peat deposits were recently discovered in the Congo?

2, 22: I think it may be more accurate to say "Prior to this conversion, drainage canals were installed" since I think the draining was largely completed as part of the "mega rice project" before the massive emissions of the 82-83 El Nino. Interesting side note, the drainage canals were built at the recommendation of a USFS expert despite the same type of drainage causing an environmental smoke disaster in the US in the 1930s as described in "Sand County Almanac."

2, 27: surface fires are not typically extinguished in Indonesia, but just burn out.

3, 1-2: Here and throughout please be careful to use "peatland" to indicate all the vegetation and peat in areas underlain by peat or "peat" to indicate just "peat". These statements should be phrased more quantitatively similar to what is presented later in the text

3, 10: "water content" of peat may be replaced by something about the "water table", i.e. peat can burn down to the groundwater level

3, 17-19: The vast majority of field data on gases emitted by smoldering peat were measured by Stockwell et al 2016 not Hu et al. who review a small subset of the gases measured. The Roulston et al paper never actually recommended any EF, they just said the instantaneous EF change during the fire. The gravimetric measurement of EFPM by Jayarathne et al., 2018 is likely the best value for reasons described later in review. Stockwell et al 2014 gave the first detailed discussion of variation in peat fire EF (Table 3 and Fig. 7) and Stockwell et al 2015 provide the most comprehensive EF data for peat fires.

Stockwell, C. E., Yokelson, R. J., Kreidenweis, S. M., Robinson, A. L., DeMott, P. J., Sullivan, R. C., Reardon, J., Ryan, K. C., Griffith, D. W. T., and Stevens, L.: Trace gas emissions from combustion of peat, crop residue, domestic biofuels, grasses, and other fuels: configuration and Fourier transform infrared (FTIR) component of the fourth Fire Lab at Missoula Experiment (FLAME-4), Atmos. Chem. Phys., 14, 9727-9754, doi:10.5194/acp-14-9727-2014, 2014.

Stockwell, C. E., Veres, P. R., Williams, J., and Yokelson, R. J.: Characterization of biomass burning emissions from cooking fires, peat, crop residue, and other fuels with high-resolution proton-transfer-reaction time-of-flight mass spectrometry, Atmos. Chem. Phys., 15, 845-865, doi:10.5194/acp-15-845-2015, 2015.

The statement "Roulston et al. (2018) and Wooster et al. (2018) found that assumed EFs for tropical peat fires could be underestimated by a factor of three" should be more quantitative. One can assume peat fire EFs are too big, or too small, by as much as they want. Roulston only said that the EF can change during the fire, but did not suggest any procedure that was more valid than random sampling. Wooster et al had experimental issues and their "peat EF" had variable contributions from non-peat fuels in all but one location as clarified more later.

3, 20: "EFs were"

3, 21-22: Kuwata et al would likely be "peatland" fires not "peat" fires since they monitored from a great distance. The idea of being able to measure a precise annual average EF implied here is probably not realistic. It's hard to measure CO2 from fires at a distance due to mixing effects and CO2 is the largest contributor to the EF. It's interesting that 13 and 19 average to 16 which is close to the 17 that Jayarathne et al essentially got. The year to year differences could be due to a number of factors besides the year. The uncertainties seem too low at e.g. 19 +/- 2.

3, 2-27: I don't think the observations were scaled so rewrite: "Previous studies have scaled particulate fire emissions from global fire emission datasets, or simulated fire-derived aerosol optical depth (AOD) or PM, by a factor 1.36 – 3.00 in order to match observations" as "Previous studies have scaled particulate fire emissions from global fire emissions inventories by a factor 1.36 – 3.00 in order to match observations of PM or simulated fire-derived aerosol optical depth (AOD)"

3, 29: "datasets" could be many things (including the papers with high EF) so change "datasets" to "inventories" throughout.

3, 33: lowering the water table exposes more peat to the possibility of burning.

3, 34: The two biggest El-Nino peat fire events were 82-83 and 97-98. The biggest El-Ninos create fire episodes that span two years.

4, 2-4: rewrite to clarify that Jayarathne is peat only and the others are for peatland.

4, 4-6: It's very important to be precise about mortality before the atmospheric community loses credibility. Smoke is a contributing factor to premature mortality, which is not like a perfectly healthy person "getting shot."

"responsible for" should be "contributed to" and "excess deaths" should be "premature deaths" so phrase reads: "contributed to between 6,513 and 17,270 premature …" and fix line 5 same way.

4, 10-14: change dataset to inventory in 4 places?

4, 30: Does dynamics and processes include evaporation and secondary formation, deposition, etc?

5, 6: POA volatility? SOA scheme? Relative amount of modeled secondary to primary emissions?

5, 16: burned per pixel

5, 25: EFs come from the peat itself? They are based on some measurement, probably the one sample of Indonesian peat burned in a lab at the time Akagi 2011 was compiled. We started working towards peat fire field measurements in 1994 against large odds. In 2000 we were able to legally import and burn one sample in our lab. In 2015 we finally completed field measurements.

6, 8: The surface fuels likely burn on the day of detection, but, if the peat ignites, having it release emissions over the next few days at least would be more realistic, but probably too late to change. Maybe it should be noted that this could shift the emissions forward in time. Also if a fire creeps along and is visible for many days hopefully it is counted as a fire all those days and not tossed as a duplicate on some days. (Later noticed this was partially addressed perhaps belatedly?)

6, 10: delete "a"

6, 11: I think hotspot detection is widely believed to be more sensitive than burned area detection so I would instead expect many hotspots did not have an associated burned area.

Can any attempt be made to use LandSat or SAR burned area data, which may be more sensitive burned area products?

6, 14: Ok, our crew in Indonesia noted surface areas that burned where the peat did not ignite by the end of the dry season so peat area burned could be less than surface area burned.

6, 15-17: Stockwell et al., (2016) measured burn depth on numerous sites (see their supplement) and report a close average value of 34 +/- 12 cm. Might be good to add to Table 1 and reference string for completeness. Similarly, Konecny et al., (2016) reported a peat bulk density of 0:120+/- 0:005 g/cm that could also go in Table 1?

6, 22-30 and Table 2: I have a number of comments on the EF in Table 2. I'll go in order by study.

Christian et al., (2003): This was just one sample of Sumatran peat burned in lab in 2000. One might be tempted to weight this study less than the more extensive studies and especially the extensive 2015 field studies of Stockwell and Smith.

Wooster et al., (2018): In general, it's important to clarify peat or peatland throughout the paper as noted above. It seems like one should not use peatland EFs for peat because FINN has EF and fuel consumption for surface vegetation already so applying peatland EF to peat will double-count? I have enough concerns about the EF in the Wooster study that it's a bit of a challenge to determine the best explanation.

1. When using light-scattering to measure PM, the response depends on particle density as they noted, but also the size distribution. They took a good step to calibrate the light-scattering with

filters also collected during the measurements. However, the graph shows many calibration points being collected at extremely high concentrations (e.g. 17-25 mg/m3 when .3-1 mg/m3 is already stretching normal measurement level). That means some PM data could be affected by unrepresentative gas to particle partitioning and greatly inflate some EFPM. This doesn't seem to have impacted their "pure peat results" from "location 5" (vide infra), but may be why they got EFPM for smoke that reflected some burning surface vegetation that were in their words "among the highest ever recorded" and well above the literature average for various surface vegetation types.

2. They use a significantly higher %C for peat than the other studies, which increases all EF across the board without saying how many peat samples were measured or how. It would have been nice to know if there was one or many peat samples (see comment on Stockwell 2015).

3. They left the carbon in the PM out of the carbon mass balance in their EF calculations. Along with ignoring NMOG this would inflate all EF by ~5%. In addition, in a unique, unexplained step; they report two different EFPM for every site depending on using CO or $CO_2$ as a reference.

4. They report EF for (1) "peatland" or "mean of all locations"; (2) "peat only" which their text says actually is "mostly peat" and includes some burning surface vegetation; (3) "really pure peat only" (location 5). For PM there are two sets of EFPM for each of these categories. In their table the EFPM range from 34.4 to 11.4. After reducing the EF for location 5 (actual pure peat) by 5% (to better reflect true total carbon emissions) the EFs are $CO_2$ (1627), CO (308), $CH_4$ (4.96), PM-avg (12.6). If any peat fire EF data were to be used from this study, then these seem appropriate and not the values in the table. In particular, I'm not sure where the 28 for EFPM the authors quote came from.

Stockwell et al 2016: Stockwell and Jayarathne co-deployed in Borneo. Stockwell reported a subset of the filter-based EF data to interpret simultaneously-collected PAX data. Thus the Stockwell OC and PM should not be included, but the BC can be since it is an independent measurement by PAX rather than thermal-optical analysis. This Stockwell study may be most extensive high quality field measurements of pure peat emissions for BC and trace gases.

Stockwell et al., 2015: This was a lab study, not a field study. It's probably not included in the paper, but nine samples of Indonesian peat measured in this study ranged from 50 to 61% C and from 50 to 58 %C when nominally at the same site; illustrating the heterogeneity and the need for intensive sampling of this parameter. The value 0.57 +/- 0.025, based on the 7 most representative samples, was used in this study, Stockwell 2016, and Jayarathne 2018.

Jayarathne et al., 2018: Gravimetric filter sampling has challenges, but is often considered the most accurate approach for PM, OC, EC, etc EFs. OC/PM ratios are best measured on the same sample.

Table 2 general comment. I think it would be best not to recycle any confusion regarding Jayarathne/Stockwell overlap and peatland versus peat. If that meant that the real best EF were different from what is shown as the Table 2 average and used in the model; and the model could not be run again, that would be understandable, but then the discussion could speculate that secondary PM or biomass burned was actually higher or add other topics.

7, 5: Stockwell et al., (2016) showed brown carbon absorption is very important for peat smoke.

7, 11 – 9, 3: This is an excellent idea to vary burn depth with guidance based on soil moisture – though water table may be even better (Putra et al., 2018). Did the soil moisture in high fire regions get significantly lower than the near constant regional average reported?

Perhaps in a future version it can be added that burning that deep can take time and that there is a lag between rainfall and water table level.

Putra, E.I., M.A. Cochrane, Y. Vetrita, L. Graham and B.H. Saharjo. 2018. Determining critical groundwater level to prevent degraded peatland from severe peat fire. IOP Conference Series: Earth and Environmental Science 149 012027 doi: 10.1088/1755-1315/149/1/012027

Also, in early November at precisely-located sites in Kalimantan, Stockwell et al 2016 (supplement) report actual burn depths for site specific comparison; or one might compare to the average of 34 cm. However, their data may be biased to dry spots that were still burning? As reported in Stockwell reaching the maximum depth takes time and termination of the fires likely requires the water table to rise since peat is difficult to "rewet".

9, 5: Excellent feature to experiment with injection altitude.

9, 8: Smoldering peat can release smoke very close to the ground (pics in Stockwell 2016 supplement).

9, 15: These are two excellent choices for plume rise experimentation in model.

9, 22 and 11, 3, Figure 1, Table 3: There is more than one PM monitoring site in Indonesia. The government agency BMKG has seven PM10 sites (http://www.bmkg.go.id/kualitas-udara/informasi-partikulat-pm10.bmkg) and 5 PM2.5 sites: (http://www.bmkg.go.id/kualitas-udara/informasi-partikulat-pm25.bmkg). Include a sentence to mention this and why not used (no access?)? I have 17 days of PM10 from 2015 for Palangkaraya that I happened to save on my hard-drive and it may be possible to get the rest. Indonesian sites could have high value for constraining emissions by virtue of being less sensitive to secondary processes, meteorology, and other sources. They would have high S:N. I also have Oct 2015, Palangkaraya visibility, which may be useful.

Any estimate of how burning in peatlands in "West Papua" might impact model/ambient-PM agreement?

9, 31: PM1/PM2.5 seems high, usually about 80%, and that's what they saw on next page.

11, 11-12: Why comparing to two months instead of whole season?

11, 14: Should it be explained somewhere that Wooster et al based estimates on an inversion using MOPPIT CO (and give MOPPIT version since last version supposedly has improved sensitivity in boundary layer, but still a lot of uncertainty)?

11, 15: "reasonable agreement"

11, 24: Jayarathne was estimating peat only. FINN + Jayarathne peat is 900 Tg right between FINNpeat and FINNpeatSM.

11, 31: "due mainly" - other things vary too, like burned area and EFPM for surface vegetation types? And should "peat" be "peatland"

12, 4: Does "smaller area" indicate that the burned area of Whitburn et al 2016 was smaller than the burned area in FINN and GFED? In general, maybe the peat consumption and other data in Whitburn et al could be compared to the products developed in this study.

Whitburn, S., Van Damme, M., Clarisse, L., Turquety, S., Clerbaux, C., and Coheur, P. F.: Doubling of Annual Ammonia Emissions from the Peat Fires in Indonesia During the 2015 El Niño, Geophys. Res. Lett., 43, 11007–11014, https://doi.org/10.1002/2016GL070620, 2016.

12, 5: "emission for peat only was" – Jayarathne et al estimated only the peat and their peat only value for PM is right between FINNpeat and FINNpeatSM peat only PM emissions. Their uncertainty is large, but properly propagated and useful since no other uncertainties are reported:)

12, 12 or later in paper: There are some more studies to compare to and it should be clear what geographic area the peat/PM ratios represent in each study. Per his short comment, Eck estimated 80-85% of aerosol was from peat combustion at Palangkaraya where the peat contribution is probably higher than in Singapore.

Eck cites Kaiser et al who got 83% of emissions from peat and Wiggins et al who got 85% of emissions from peat.

Kaiser, J. W., van der Werf, G. R., & Heil, A. (2016). Global climate biomass burning in "State of the climate in 2015". Bulletin of the American Meteorological Society, 97(8), S1–S275. https://doi.org/10.1175/2016BAMSStateoftheClimate.1

Engling et al., 2014 got 76% peat fire impact on TSP at Singapore for hazy days in 2006.

Engling, G., He, J., Betha, R., and Balasubramanian, R.: Assessing the regional impact of Indonesian biomass burning emissions based on organic molecular tracers and chemical mass

balance modeling, Atmos. Chem. Phys., 14, 8043-8054, https://doi.org/10.5194/acp-14-8043-2014, 2014.

Possibly useful?

Hansen, A. B., Witham, C. S., Chong, W. M., Kendall, E., Chew, B. N., Gan, C., Hort, M. C., and Lee, S.-Y.: Haze in Singapore – source attribution of biomass burning PM10 from Southeast Asia, Atmos. Chem. Phys., 19, 5363-5385, https://doi.org/10.5194/acp-19-5363-2019, 2019.

13, 5-9: How much do emissions from the island of New Guinea impact the monitoring stations employed in this study (see Fig. 2 in Hansen et al above)? Also, line 9, "need for future"

13, 18: The word "matches" is used here and elsewhere when maybe "consistent with" is better or a % difference would be more useful.

13, 19-25: Would it help if the model released the peat emissions over the weeks following detection? Peak regional PM in Palangkaraya was in October rather than September.

Figure 4: FINNpeat and FINNpeatSM seem to overestimate PM and increasingly so with distance in the two examples shown. One could argue that the sites closer to the fires and with higher PM are best for constraining initial emissions or EFs for PM. Downwind sites are more subject to secondary formation, evaporation, deposition, or transport errors. However, Fig. 5 does show the lowest bias overall and the highest correlation overall for FINNpeatSM, which is useful. Is there value in comparing the integrated amount of PM modeled vs measured over the season (or by month) by site in a Table? Is there any CO monitoring to compare to?

14, 5-8: Surface fuels burn quickly after detection and then the peat smolders till the end of the dry season. At the end of the dry season we saw an average burn depth of 34 cm, but burning down to that depth and or laterally across much of the burn scar takes time. The model choice to shift all peat emissions forward to the day of detection could cause Sept/Oct too high? On lines 6 and 7, be specific using "model" or "models", all of them, or which ones?

14, 13-16: The effect of a non-linear relationship might also be mimicked by a longer time-frame for peat consumption as speculated above.

14, 16: Has a relationship between soil moisture and burn depth in fact been measured? The Putra et al reference cited above discusses a large data base relating rainfall, water table, and burn depth.

14, 18-20: This seems to finally acknowledge the point about the emissions taking time.

15, 1-2: "Only hotspots that last 3 days are a fire" makes no sense so it's good the authors ignored this. There could be cases where a 3-day fire is needed to ignite wetter peat.

15, 3: The timing of the peaks in Pekanbaru is excellent.

16, 5: Without knowing anything about secondary processes the best way to validate EF is with EF measurements and the authors are using updated EF, but undetected burned area can't really be written off? There could be undetected fires that are offset because the model could e.g. underestimate evaporation?

17, 1-2: Injection throughout the boundary layer, as opposed to all at the surface always reduces the modeled PM. Per above, there could be missing fires that are compensated for in the injection choice. Is there any model-assumed evaporation associated with faster dilution into the whole boundary layer?

17, 12: Is POA inert in model? A recent study suggests about 50% of PM can be lost by evaporation (or deposition) for some fires.

Selimovic, V., Yokelson, R. J., McMeeking, G. R., and Coefield, S.: In situ measurements of trace gases, PM, and aerosol optical properties during the 2017 NW US wildfire smoke event, Atmos. Chem. Phys., 19, 3905-3926, https://doi.org/10.5194/acp-19-3905-2019, 2019.

Other BB studies (e.g. Vakkari et al., 2014; 2018) saw SOA dominate.

17, 23-24: The author's model gets a better r value with AOD, not sure if this sentence is needed?

17, 29: "mean" should be "peak"? Otherwise a mean of 1800 seems to conflict with the much lower means given just below?

18, 1-6: Are these peat/PM values for whole model domain or some subset? The values are higher than for the study of Lee et al discussed on next page

18, 6: So does this discussion then indicate that PM/CO increases slightly with aging. E.g. a factor of 76/71 still fairly close to the fires? Also, is this really known to be all from SOA when there are also inorganic precursors like NH3, NOx, SO2, etc?

18, 11-12: don't understand this.

19, 3: Does current study agree that half or more of the pollution during haze episodes in Singapore is not from fires?

19, 4-6: not sure what this adds?

19, 8-20, 2: More sites close to the fires (like Palangkaraya) could help determine the proper choice of injection altitude? Maybe it makes sense to inject non-peat emissions throughout the BL and peat fire emissions at the surface?

20, 7: In the AOD calculation, is the air-mass factor impacted at some PM level, is BrC ignored at 550, is it possible to compare to inferred AOD at Palangkaraya in Eck et al?

20, 11: add "surface" before "PM2.5" to make this conclusion even more obvious?

20, 16: Here and earlier, a more inclusive geographic term than "Indonesia" might be "insular Southeast Asia."

21,1: "datasets" to "inventories"? If not specifying inventories, should the qualifier "some" be included since there is agreement with some previous work?

21, 3: This study simulated more than just Sept-Oct so why limit to those months?

21, 8: fix "a new emissions datasets".

The new EF were likely applied to peat fires in Malaysia too? There can be many peat fires in Sarawak (one part of Malaysian Borneo). The Smith et al study measured peat fire EF on mainland Malaysia. Also lines 26, 28, 32 may overly single out Indonesia, which dominates but doesn't account for all the peat combustion in the model domain.

21,12: mention agreement with some other estimates?

21, 13: improved estimates of peat consumption should improve $CO_2$ as well?

21, 19: Need work on secondary processes also.

21, 24: Work is needed on how rainfall, soil moisture, water table depth, burn depth, and burn rate or timing are related. In Putra et al TRMM rainfall had a minimum in August, but measured ground water level in an extensive series of dip wells hit a minimum one or two months later.

21, 31: "impact" better as "contribution" since the ambient PM itself was not revised.

---

## Referee Comment (RC2) · Anonymous Referee #2 · 21 Jun 2019

**Review of "Revised estimate of particulate emissions from Indonesian peat fires in 2015," by Kiely et al.**

This paper revisits estimates of smoke emissions from the Indonesian fires of 2015. As is well known, fire emission inventories are highly uncertain, with large discrepancies among them. Here the authors take the FINN emissions as a base case, and then attempt to improve these emissions. Fire emissions from peatland is expected to be a large contributor to smoke in this region, but the standard FINN inventory does not consider emissions from peatland. In addition, the depth of peat burned, a quantity governed by soil moisture, may influence the magnitude of smoke emissions in this region. To address these weaknesses in the FINN inventory, the authors do the following sensitivity studies: (1) adding the peat emissions from the GFED inventory to FINN, (2) recreating peat emissions with land cover maps, and (3) using remotely sensed soil moisture to constrain the magnitude of peat burned. While the base FINN simulation underestimates observations, all three efforts listed above improve the model match with observed surface PM and with aerosol optical depths (AOD). Sensitivity study #3 comes the closest to observations. The authors estimate that vegetation and peat fires increased $PM_{2.5}$ concentrations over Sumatra and Borneo during September and October 2015 by $> 125$ $\mu g$ $m^{-3}$. Such a heavy particulate burden has large significance for human and ecosystem health.

The paper is excellent, in my view, and should be published after the following comments are addressed.

**Main comment.**
On page 16, the authors claim that PM emissions in GFED4 are underestimated because emission factors for peat combustion are too small. In my view, the authors cannot so easily dismiss the large role of clouds and haze in these or any smoke underestimates. GFED4 relies in part on area burned in constructing emission inventories, as well as on active fire information. FINN relies only on active fire information. Both kinds of data can be obscured by clouds and haze. (See for example Kaiser et al., 2012.) The authors should be aware that the interference from clouds and haze can vary strongly from month to month and from fire to fire, depending on meteorology and the magnitude of the fire itself. Cusworth et al. (2018) found that emissions from agricultural fires in India were more strongly underestimated during high fire years. At least two fire emission inventories, GFAS and QFED, make cloud-gap adjustments to account for fires obscured by clouds/haze.

Indeed the variation of the cloud/haze interference could explain the inability of simulations in this work to capture the lack of temporal trend from September to October in the observations. The observations show relatively similar values of surface $PM_{2.5}$ and AOD in both months (Figures 5b and 6b), while all three sensitivity simulations show as much as double the value in September as in October. Applying GFAS emissions to their model, Koplitz et al. (2015) captured the observed lack of trend from September to October in monthly mean $PM_{2.5}$ values over Singapore.

**Minor comments.**

Page 5. The text says that the model meteorology is allowed to run freely through the month, and then is reinitialized at the beginning of the month. Is that right? Would this approach lead to a discontinuity in meteorology at monthly intervals? Maybe a reference would be helpful.

Page 5. In the FINN+GFEDpeat simulation, is there concern that the inventory double-counts emissions? In this inventory, the GFED peat emissions are added to the FINN vegetation emissions.

Page 6. Citing Tansey et al. (2008), the authors state that "60% of burned areas did not have an identified hotspot, implying an area burned per MODIS hotspot of approximately 40 ha." Are these fires in Indonesia only?

Page 9. The text states, "Fires can inject emissions above the surface." The author should clarify that they mean the surface layer of air, which is xx meters in the model.

Page 11. What exactly comprises $PM_{2.5}$ in the model? Is this mainly BC and OC?

Page 12. The authors should consider including emissions of BC and OC in Table 4.

Page 13, Figure 2. The plots would be easier to interpret if low values were colored white.

Page 14, Figure 3. The caption should state what region is shown.

Page 15, Figure 4. The caption should provide the temporal correlation of modeled values and measurements for these two sites.

Page 15. The text states, "The overestimation in September could also be due to an issue with fire detection. Syaufina and Sitanggang (2018) found that only hotspots which last for 3 days indicate fires, something which is not considered when calculating the emissions." This doesn't seem right. Agricultural fires in particular may last just hours. Indeed, a problem with active fire detection is that the satellite overpass time may miss a short-lived fire, giving rise to the phenomenon that the authors note on page 6, with some burned areas not associated with any hotspot. In any event, if a hotspot lasts less than 3 days and is not a fire, what else could it be?

Page 18, Figure 6b. Wrong unit is given.

Page 19, Figure 7. Again the authors should consider using a white color for very low values so that other colors stand out.

Page 19, Figure 8. Colors in color bar are hard to see. Bar should be fatter.

Page 20, Line 11. "Effected" should be "affected."

Pages 20-21, Conclusions. The authors should consider briefly discussing the merits of other emission inventories used to simulate agricultural fires in Indonesia (e.g., GFAS).

Supplement page 3, Figure S3. Caption should state region shown. Also some statistics should be provided regarding the match between model and observations.

Supplement page 4-5, Figure S5-S6. Comment same as for Figure 7.

**References.**

Cusworth, D.H., et al., Quantifying the influence of agricultural fires in northwest India on urban air pollution in Delhi, India, Env. Res. Let., 13, 2018.

Kaiser, J.W., et al., Biomass burning emissions estimated with a global fire assimilation system based on observed fire radiative power, Biogeosciences, 9, 527–554, 2012.

Koplitz, S. N., et al., Public health impacts of the severe haze in Equatorial Asia in September-October 2015: Demonstration of a new framework for informing fire management strategies to reduce downwind smoke exposure, Environ. Res. Lett., 11, 094023, 2016.

---

## Author Response (AR1)

**New estimate of particulate emissions from Indonesian peat fires in 2015**

[revised manuscript text omitted]

20 HCN, $NH_3$ and PM (Stockwell et al., 2016). Until recently there have been few specific measurements of EFs for tropical peat fires. Roulston et al. (2018) and Wooster et al. (2018) found that  EFs for tropical peat fires could be underestimated by a factor of three ($PM_{2.5}$ EF from peat fires is assumed to be 9.1 g $kg^{-1}$ in GFED4, compared to 24 g $kg^{-1}$ suggested by Roulston et al. (2018) and 28 g $kg^{-1}$ suggested by Wooster et al. (2018)). There are large variations in EFs for peat in Indonesia. In one study measuring emissions from peat fires in Central Kalimantan during 7 days in 2015, $PM_{2.5}$ EFs were found to

25 vary between 6 and 30 g $kg^{-1}$ (Jayarathne et al., 2018).  Kuwata et al. (2018) used measurements from Indonesian peatland fires to estimate EFs of $PM_{10}$ of 13±2 g $kg^{-1}$ in 2013 and 19±2 g $kg^{-1}$ in 2014.

These uncertainties cause corresponding uncertainty in estimates of emissions from peat fires, and impacts on the regional air

30 pollution. Previous studies underestimate measured and aerosol optical depth (AOD) or PM, and scale particulate fire e  (Marlier et al., 2012; Ward et al., 2012; Johnston et al., 2012;

Tosca et al., 2013; Reddington et al., 2016; Koplitz et al., 2016). This suggests that particulate emissions from tropical peatland regions are underestimated in current fire emission inventories.

Severe fire events in Indonesia occur during periods of drought (van der Werf et al., 2008; Tosca et al., 2011; Gaveau et al., 2014; Field et al., 2016), resulting in strong seasonal and interannual variability. Severe droughts lower the water table,  exposing more peat and increasing the susceptibility of burning.  Extensive fires and regional haze episodes across Indonesia have occurred in 1982-1983, 1997-1998, 2006, 2009, 2013 and 2015. During September to October 2015, dry conditions caused by a strong El Niño, resulted in large fires across Sumatra and Kalimantan. This fire episode was the largest in Indonesia since 1997 (Huijnen et al., 2016), releasing an estimated  188±67 TgC (Huijnen et al., 2016) as $CO_2$ and 149±71 TgC from peat fires (Jayarathne et al., 2018). The fires also emitted substantial amounts of $PM_{2.5}$ estimated at  9.1±3.2 Tg (Wooster et al., 2018), with 6.5±5.5 Tg from peat fires (Jayarathne et al., 2018). Particulate air pollution from these fires may have caused  between 6,513 and 17,270 excess premature deaths through short term exposure to fire-sourced $PM_{2.5}$ (Crippa et al., 2016) and as many as 100,300 excess premature deaths over the longer term due to exposure to this pollution (Koplitz et al., 2016).

[revised manuscript text omitted]

estimates greater dry matter consumption. (Whitburn et al., (2016), have estimated dry matter fuel consumption of 525 Tg, calculated using satellite CO emissions (from the IASI instrument) from peatlands, and the CO EF for peat from GFAS. This estimate is larger than that found for GFED or FINNpeatSM, and smaller than that found for FINNpeat.

5 Total September and October 2015 emissions of $CO_2$ follow a similar pattern to dry matter consumption, with similar values for GFED, FINN+GFEDpeat and FINNpeatSM (773 Tg, 822 Tg and 781 Tg), largest emissions for FINNpeat (1014 Tg) and smallest emissions for FINN (353 Tg). $CO_2$ EFs, are similar for GFED and FINN peat (1703 g kg $^{-1}$ and 1669 g kg $^{-1}$), explaining the similarity between dry matter consumption and $CO_2$ emissions for these datasetsinventories. The total $CO_2$ emissions for September to October estimated by Wooster et al., (2018) was 692±213 Tg, matching GFED, FINN+GFEDpeat

10 and FINNpeatSM. Jayarathne et al., (2018) estimated 547±259 Tg of $CO_2$ were emitted from peat fires over South Sumatra and Kalimantan, a range which includes the total $CO_2$ emissions from peat fires for FINN+GFEDpeat (469 Tg), FINNpeat (661 Tg) and FINNpeatSM (428 Tg).suggesting that the total $CO_2$ emissions from FINN (353 Tg for Borneo and Sumatra) are too small, due to lack of peat fires in this dataset.

15 Total emissions of $PM_{2.5}$ vary across simulations due to differences in assumed $PM_{2.5}$ EFs. FINN has the smallest total $PM_{2.5}$ emissions for September to October (2.09 Tg; Table 4). GFED and FINN+GFEDpeat have similar total $PM_{2.5}$ emissions (4.14 Tg and 4.60 Tg), smaller than that for FINNpeatSM (7.33 Tg) despite these datasetsinventories having similar dry matter consumption and $CO_2$ emissions. This is due mainly to the difference in the assumed EFs for $PM_{2.5}$ from peat fires, with 9.1 g kg $^{-1}$ used in GFED,

20 and 22.26 g kg $^{-1}$ used in FINNpeatSM. Wooster et al., (2018), assumed a $PM_{2.5}$ EF of 28±6 g kg $^{-1}$ and estimated that 9.1±3.2 Tg of $PM_{2.5}$ was emitted over the whole of Sumatra and Kalimantan for September and October 2015, similar to that found in FINNpeat (10.60 Tg) and FINNpeatSM (7.33 Tg). In contrast, FINN and FINN+GFED, which use the lower EF, produce smaller $PM_{2.5}$ emissions by a factor of 2 and 4 respectively. Jayarathne et al. (2018) found that, for a smaller study area inof Sumatra and Kalimantan, the total $PM_{2.5}$ emission from peat fires was 6±5.5 Tg, a range which covers the total $PM_{2.5}$ emissions

25 from peat fires from FINN+GFEDpeat (2.51 Tg), FINNpeat (8.51 Tg) and FINNpeatSM (5.24 Tg)all of the datasets.

GFED, FINN+GFED and FINNpeatSM all emit similar total amounts of CO over September and October (75 Tg, 77 Tg and 78 Tg respectively). This is likely due to the similar EFs used for peat fires (243 g kg $^{-1}$ in FINNpeatSM, 210 g kg $^{-1}$ in GFED). The total CO emissions from FINN are smaller (20 Tg) and from FINNpeat are larger (109 Tg). The contribution from

30 peat fires to total CO emissions (74 - 82%) is larger than for $CO_2$ (55 – 65%) and $PM_{2.5}$ (55% - 80%). For every 1g of CO emitted from fires, 0.04g of SOA is assumed, and the total SOA from each fire emission inventory is shown in Table 4. The SOA increases the total PM emitted from fires to 2.89 Tg from FINN, 7.14 Tg from GFED, 7.68 Tg from FINN+GFED, 14.96 Tg from FINNpeat and 10.45 Tg from FINNpeatSM.

Table 4 also shows the fraction of  emissions that are estimated to come from peat fires. Across the inventories that include peat burning, peat fires contribute 51-62% of dry matter consumption, 55-65% of $CO_2$ emissions and 55-80% of $PM_{2.5}$ emissions. The emission inventories with updated $PM_{2.5}$ emission factors result in a greater contribution from peat burning (71%-80%) compared to emission inventories with the older EFs (55%-62%).

5 Wooster et al. (2018) found that peat fires contributed 85% of the dry matter fuel consumption, and 95% of the $PM_{2.5}$ emissions in September and October 2015,  greater than our estimates with updated EFs.

Figure 2 shows the spatial variations of the total $PM_{2.5}$ emissions during September and October 2015. In all inventories, greatest emissions occur in southern Kalimantan and central and southern Sumatra, matching the locations

10 of peatlands (Fig. 1). For the FINNpeatSM emissions, Sumatra contributes 42% of the total $PM_{2.5}$ emissions, for FINNpeat, FINN+GFEDpeat and FINN, the contribution is 39%, 40% and 32% respectively. Wooster et al. (2018) found that 33% of the total $PM_{2.5}$ emissions came from Sumatra, while Koplitz et al. (2016) found that that 47% of OC and BC emitted in June to October 2015 came from Sumatra. Our estimates exclude fire emissions from eastern Indonesia. Nechita-banda et al. (2018) estimated that fires in eastern Indonesia and Papua New Guinea contributed around 15-20% of total CO emissions from fires

15 across the region, highlighting the need future work to quantify PM emissions in this region.

FINN and GFED underestimate total emitted $PM_{2.5}$ and FINN underestimates total emitted $CO_2$ compared to the emissions found by Wooster et al. (2018) and Jayarathne et al. (2018), suggesting that peat fires are important contributors to these emissions. FINNpeatSM is the only emissions inventory that is consistent with  these previous studies for both $PM_{2.5}$ and $CO_2$.

[Figure]

**Figure 2: Total PM₂.₅ fire emissions during September-October 2015 (g /m2).**

Figure 3 shows daily total $PM_{2.5}$ emissions from the different inventories over the study area. Temporally, the inventories follow a similar pattern, with 80-90% of the total $PM_{2.5}$ emissions for 2015 occurring in August-October. For all the emissions inventories the majority of emissions are in September, followed by October and then August. GFED has the largest difference between September and October emissions (58% in Sep and 17% in Oct), followed by FINN+GFEDpeat (47% and 24%), FINNpeat (36% and 30%), and finally FINN (33% and 29%) and

[Figure]

**Figure 3: Total daily PM2.5 emissions from fires during 2015. Total shown for the area in Figure 1, 95-120°E and 10°S-10°N.**

FINNpeatSM (36% and 32%) which have the smallest differences between the two months. The reduced ratio of the fraction of emissions in September compared to October for FINNpeatSM is due to greater soil moisture in September resulting in a reduced peat burn depth.

5  Another commonly used emissions inventory is the Global Fire Assimilation System (GFAS1). This uses satellite fire radiative power to detect fires, combined with EFs to calculate daily emissions (Kaiser et al., 2012). For peat fires, some EFs are from studies of Indonesian peat, although the PM2.5 EF (9.1 g/kg) is from tropical vegetation, as is used in GFED. Reddington et al. found that GFAS1 requires the same scaling as GFEDv3 to match observations in Indonesia. It is therefore likely that GFAS1 would show similar results to GFED in our assessment.

**3.2 Comparison of model and observational data**

We evaluated the WRF-chem simulations with the different emissions inventories and injection options against measured $PM_{10}$, $PM_{2.5}$ and $PM_1$ concentrations.

Figures 4 and 5 shows the comparison of simulated and observed PM concentrations. Comparisons of $PM_{2.5}$ and $PM_1$
15   measurements only, which were restricted to Singapore, are shown in Fig. S3.

PM concentrations are underestimated by the model with FINN emissions, with a fractional bias (FB) of -0.67 with surface injection and -0.77 with boundary layer injection of emissions, with an average across both simulations of -0.72. The model with FINN+GFEDpeat emissions also underestimates PM concentrations (average FB= -0.35) whilst the model with FINNpeat

[Figure]

**Figure 4: Daily observed and modelled (a) PM$_{2.5}$ in Singapore, and (b) PM$_{10}$ in Pekanbaru, for WRF-chem runs with different fire emissions inventories and the surface injection option. a) shows observations of PM$_{2.5}$ (solid) and PM$_1$ (dashed). The Pearson's correlation (r) for (a) is 0.47, 0.73, 0.52 and 0.50, and for (b) is 0.63, 0.60, 0.65 and 0.73 for FINN, FINN+GFEDpeat, FINNpeat and FINNpeatSM respectively.**

emissions overestimates PM concentrations (average FB=0.2). The model with FINNpeatSM emissions has the smallest bias

(average FB=-0.11, suggesting mean emissions from this inventory are closest to reality.

[revised manuscript text omitted]

Singapore was from peat fires, slightly higher than the contribution of peat fires to the simulated PM$_{2.5}$ concentration shown

15 in Figure 8 (67%). (Engling et al., (2014) found that in 2006, 76% of particulate matter in Singapore was from peat fires. At the Palangkaraya AERONET site in Kalimantan, (Eck et al., (2019) found that 80-85% of AOD came from peat burning, consistent with the contribution of peat fires to PM$_{2.5}$ simulated by the model at that location (Figure 8)

Inclusion of emissions from peat fires gives largest fire emissions in Sumatra and western Kalimantan, where previously

20 emissions were substantially lower (Fig. 2). This leads to higher PM$_{2.5}$ concentrations across Singapore (Fig. 5), which has a large impact on the population exposure to the PM$_{2.5}$. The location of fires can be an important factor of their contribution to air pollution in populated areas. Lee et al. (2017) found that the 2015 fires in Sumatra accounted for 50% of fire-derived PM$_{2.5}$ in Kuala Lumpur and 41% in Singapore, and (Hansen et al., (2018), found that during August to October fires in South Sumatra and Central Kalimantan are the largest contributors to PM$_{2.5}$ in Singapore. . Reddington et al. (2014) and Kim et al. (2015)

[revised manuscript text omitted]

Eck, T.F., Holben, B.N., Giles, D.M., Slutsker, I., Sinyuk, A., Schafer, J.S., Smirnov, A., Sorokin, M., Reid, J.S., Sayer, A.M., Hsu, N.C., Shi, Y.R., Levy, R.C., Lyapustin, A., Rahman, M.A., Liew, S.C., Salinas Cortijo, S. V., Li, T., Kalbermatter, D., Keong, K.L., Yuggotomo, M.E., Aditya, F., Mohamad, M., Mahmud, M., Chong, T.K., Lim, H.S., Choon, Y.E., Deranadyan, G., Kusumaningtyas, S.D.A. and Aldrian, E. 2019. AERONET Remotely Sensed Measurements and Retrievals of Biomass Burning Aerosol Optical Properties During the 2015 Indonesian Burning Season. *Journal of Geophysical Research: Atmospheres*. **124**(8), pp.4722–4740.

Emmons, L.K., Walters, S., Hess, P.G., Lamarque, J., Pfister, G.G., Fillmore, D., Granier, C., Guenther, A., Kinnison, D., Laepple, T., Orlando, J.J., Tie, X., Tyndall, G., Wiedinmyer, C., Baughcum, S.L. and Kloster, S. 2010. Description and evaluation of the Model for Ozone and Related chemical Tracers , version 4 ( MOZART-4 ). *Geoscientific Model Development*., pp.43–67.

Engling, G., He, J., Betha, R. and Balasubramanian, R. 2014. Assessing the regional impact of indonesian biomass burning emissions based on organic molecular tracers and chemical mass balance modeling. *Atmospheric Chemistry and Physics*. **14**(15), pp.8043–8054.

Field, R.D., van der Werf, G.R., Fanin, T., Fetzer, E.J., Fuller, R., Jethva, H., Levy, R., Livesey, N.J., Luo, M., Torres, O. and Worden, H.M. 2016. Indonesian fire activity and smoke pollution in 2015 show persistent nonlinear sensitivity to El Niño-induced drought. *Proceedings of the National Academy of Sciences*. [Online]. **113**(33), pp.9204–9209. Available from: http://www.pnas.org/lookup/doi/10.1073/pnas.1524888113.

Freitas, S.R., Longo, K.M., Chatfield, R., Latham, D., Dias, M.A.F.S., Andreae, M.O., Prins, E., Santos, J.C., Gielow, R. and Carvalho, J.A. 2007. Including the sub-grid scale plume rise of vegetation fires in low resolution atmospheric transport models. *Atmospheric Chemistry and Physics*., pp.3385–3398.

Gaveau, D.L.A., Salim, M. a, Hergoualc'H, K., Locatelli, B., Sloan, S., Wooster, M., Marlier, M.E., Molidena, E., Yaen, H., DeFries, R., Verchot, L., Murdiyarso, D., Nasi, R., Holmgren, P. and Sheil, D. 2014. Major atmospheric emissions from peat fires in Southeast Asia during non-drought years: Evidence from the 2013 Sumatran fires. *Scientific Reports*. **4**.

Gaveau, D.L.A., Sloan, S., Molidena, E., Yaen, H., Sheil, D., Abram, N.K., Ancrenaz, M., Nasi, R., Quinones, M., Wielaard, N. and Meijaard, E. 2014. Four decades of forest persistence, clearance and logging on Borneo. *PLoS ONE*. **9**(7), pp.1–11.

Ge, C., Wang, J. and Reid, J.S. 2014. Mesoscale modeling of smoke transport over the Southeast Asian Maritime Continent : coupling of smoke direct radiative effect below and above the low-level clouds. *Atmospheric Chemistry and Physics*., pp.159–174.

Giglio, L., Randerson, J.T. and Van Der Werf, G.R. 2013. Analysis of daily, monthly, and annual burned area using the fourth-generation global fire emissions database (GFED4). *Journal of Geophysical Research: Biogeosciences*. **118**(1), pp.317–328.

Grell, G.A., Peckham, S.E., Schmitz, R., Mckeen, S.A., Frost, G., Skamarock, W.C. and Eder, B. 2005. Fully coupled ''' online ''' chemistry within the WRF model. *Atmospheric Environment*. **39**, pp.6957–6975.

Gruber, A., Dorigo, W.A., Crow, W. and Wagner, W. 2017. Triple Collocation-Based Merging of Satellite Soil Moisture Retrievals. *IEEE Transactions on Geoscience and Remote Sensing*. **55**(12), pp.6780–6792.

Guenther, A., Karl, T., Harley, P., Wiedinmyer, C., Palmer, P.I. and Geron, C. 2006. Estimates of global terrestrial isoprene emissions using MEGAN ( Model of Emissions of Gases and Aerosols from Nature ). *Atmospheric Chemistry and Physics*., pp.3181–3210.

Hansen, A.B., Chong, W.M., Kendall, E., Chew, B.N., Gan, C., Hort, M.C., Lee, S.-Y. and Witham, C.S. 2018. Haze in Singapore – Source Attribution of Biomass Burning from Southeast Asia. *Atmospheric Chemistry and Physics Discussions*., pp.1–29.

Hansen, M.C.C., Potapov, P. V, Moore, R., Hancher, M., Turubanova, S.A. a, Tyukavina, A., Thau, D., Stehman, S.V. V, Goetz, S.J.J., Loveland, T.R.R., Kommareddy, A., Egorov, A., Chini, L., Justice, C.O.O. and Townshend, J.R.G.R.G. 2013. High-Resolution Global Maps of 21st-Century Forest Cover Change. *Science*. [Online]. **342**(November), pp.850–854. Available from: http://www.ncbi.nlm.nih.gov/pubmed/24233722.

Hatch, L.E., Luo, W., Pankow, J.F., Yokelson, R.J., Stockwell, C.E. and Barsanti, K.C. 2015. Identification and quantification of gaseous organic compounds emitted from biomass burning using two-dimensional gas chromatography-time-of-flight mass spectrometry. *Atmospheric Chemistry and Physics*. **15**(4), pp.1865–1899.

Heil, A., Langman, B. and Aldrian, E. 2007. Indonesian peat and vegetation fire emissions : Study on factors influencing large-scale smoke haze pollution using a regional atmospheric chemistry model Indonesian peat and vegetation fire emissions : Study on factors influencing large-scale smoke haze. *Mitigation and Adaption Stratagies for Global Change*. (January).

Hodzic, A. and Jimenez, J.L. 2011. Modeling anthropogenically controlled secondary organic aerosols in a megacity: A simplified framework for global and climate models. *Geoscientific Model Development*. **4**(4), pp.901–917.

[revised manuscript text omitted]

Whitburn, S., Damme, M. Van, Clarisse, L., Turquety, S., Clerbaux, C. and Coheur, P.-. 2016. Peat fires doubled annual ammonia emissions in Indonesia during the 2015 El Niño. *Geophysical Research Letters*.

Wiedinmyer, C., Akagi, S.K., Yokelson, R.J., Emmons, L.K., Al-Saadi, J. a., Orlando, J.J. and Soja, a. J. 2011. The Fire

INventory from NCAR (FINN) – a high resolution global model to estimate the emissions from open burning. *Geoscientific Model Development Discussions*. **3**(4), pp.2439–2476.

Wiggins, E.B., Czimczik, C.I., Santos, G.M., Chen, Y., Xu, X., Holden, S.R., Randerson, J.T., Harvey, C.F., Ming, F. and Yu, L.E. 2018. Smoke radiocarbon measurements from Indonesian fires provide evidence for burning of millennia- aged peat. *Proceedings of the National Academy of Sciences*. **115**(49), pp.12419–12424.

Wooster, M., Gaveau, D., Salim, M., Zhang, T., Xu, W., Green, D., Huijnen, V., Murdiyarso, D., Gunawan, D., Borchard, N., Schirrmann, M., Main, B. and Sepriando, A. 2018. New Tropical Peatland Gas and Particulate Emissions Factors Indicate 2015 Indonesian Fires Released Far More Particulate Matter (but Less Methane) than Current Inventories Imply. *Remote Sensing*. [Online]. **10**(4), p.495. Available from: http://www.mdpi.com/2072-4292/10/4/495.

Wösten, J.H.M., Clymans, E., Page, S.E., Rieley, J.O. and Limin, S.H. 2008. Peat – water interrelationships in a tropical peatland ecosystem in Southeast Asia. . **73**, pp.212–224.

WRI n.d. World Resources Institute. 'Peat lands'. Accessed through Global Forest Watch on [17/04/2017]. Available from: www.globalforestwatch.org.

Zaveri, R.A., Easter, R.C., Fast, J.D. and Peters, L.K. 2008. Model for Simulating Aerosol Interactions and Chemistry ( MOSAIC ). . **113**(July), pp.1–29.

**Below is a response to all comments, including all changes which have been made to the manuscript. Under the response is the complete manuscript, with all the changes shown using track changes.**

**Response to Review**
**Kiely et al**

We would like to thank the referees for taking time to review our manuscript and for all the comments they have provided. We have responded to all the referee comments below and have modified our manuscript accordingly. Our manuscript has been strongly improved through this process and we hope it is now suitable for publication. To guide the review process, referee comments below are in plain text and our responses are in blue italics text.

**Review of peat fire emissions estimate by Kiely et al.**
**Bob Yokelson**

In this work new peat fire emission factor (EF) measurements from the literature and a new approach for model estimates of peat consumption developed by the authors are incorporated into the authors WRF-chem regional model for insular SE Asia and better agreement with ambient PM monitoring and AOD are observed. The impact of injection altitude is also explored. The new peat consumption approach may be developed further, but has already reduced bias and increased correlation. A few improvements could be made to the manuscript. Mainly, a little more could be added about the impact of secondary processes and efforts made to avoid recycling some confusion on literature EFs. My suggestions are further clarified in my detailed comments provided in order of appearance in the paper in "page number, line number:" format if not otherwise specified. If, after commenting at one "P, L", I saw the issue partially addressed later in the paper, I did not delete my first comment in most cases. This is because it may be worth addressing the issue earlier in the paper?

Title: very minor point "Revised" kind of implies they are changing a previous estimate they made?

*We agree that this could be confusing. We want to maintain the message that our work is contributing to a new, revised estimate of emissions. We change the title to 'New estimate of particulate emissions from Indonesian peat fires in 2015'.*

1, 22: Two minor points. 1) There is a lot of peat in Malaysian Borneo. 2) The drainage canals may have already largely been built by the early 1980s? There's a debate now about blocking them to re-flood the peatlands.

*1) This is a good point. However, the 2015 fires were mostly in Indonesia and our paper only includes new peat emissions for Indonesian fires. This has been specified in the description of the emissions inventories:*

*'The peatland map only includes peatlands in Indonesia, so emissions from Malaysian peat fires are not included.'*

*2) We have changed 'being' to 'have been' to reflect that this has largely already occurred.*

1, 25: "fuel consumption" may be better/safer than "burn depth" since the area burned is hard to measure due to small size, smoldering, high regional cloud cover, etc. (Reid et al., 2013)

Reid, J. S., Hyer, E. J., Johnson, R., Holben, B. N., Yokelson, R. J., Zhang, J., Campbell, J. R., Christopher, S. A., Di Girolamo , L., Giglio, L., Holz, R. E., Kearney, C., Miettinen, J., Reid, E. A., Turk, F. J., Wang, J., Xian, P., Zhao, G., Balasubramanian, R., Chew, B. N., Janai, S., Lagrosas, N., Lestari, P., Lin, N.-H., Mahmud, M., Nguyen, A. X., Norris, B., Oahn, N. T. K., Oo, M., Salinas, S. V., Welton, E. J., Liew, S. C.: Observing and understanding the Southeast Asian aerosol system by remote sensing: An initial review and analysis for the Seven Southeast Asian Studies (7SEAS) program, Atmos. Res., 122, 403-468, doi:10.1016/j.atmosres.2012.06.005, 2013.

*We agree that 'fuel consumption' is better here and have changed it.*

2. 1: GFED4s, the "s" indicates the product with small fires added?
*GFED4s is the version of the product with emissions from small fires added. We have included this as,*
*'Global Fire Emissions Database (GFED4s, with small fires)'*

2, 2-3: Technically, to "confirm" an EF with a model you'd have to compare the model to the measurements at the point of emission or somehow know that the model correctly treated all secondary processes, so it's probably better to claim "consistency" or "support" for the new EF. Also the paper should mention the relative change in PM due to post emission processes predicted by the model at some point.

*We would like to thank the reviewer for this point about the specific language used, we agree that 'confirm' is not the correct term. We have changed 'provides an independent confirmation' to 'provides independent support'.*
*We have included the following in the methods section, about how post emissions processes may affect the results.*

*"Atmospheric PM concentrations are impacted by a range of atmospheric processes including atmospheric transport, deposition and secondary production of aerosol. Evaluating the fire emissions is complicated by the treatment of these processes in the model. "*

2, 19: Is this statistic still current now that large peat deposits were recently discovered in the Congo?

*The reviewer is correct that the statistic of Indonesia containing 47% of the world's tropical peat is not including peatlands discovered in the Congo, and is therefore no longer correct. Including this peatland, it is more likely to be 36% (Using 145,500km2 of peatland in the Congo*

*as estimated by* Dargie et al. (2017)*, instead of 6219km2 suggested in* Page et al. (2011)*, and recalculating the total tropical peatland in Page et al. (2011) to be 580,306km2 instead of 441,025km2).*

*The sentence has been changed to:*
*'Indonesia contains 36% of the world's tropical peatland, the largest of any country in the tropics  (Page et al., 2011; Dargie et al., 2017)'*

*Dargie, G.C., Lewis, S.L., Lawson, I.T., Mitchard, E.T.A., Page, S.E., Bocko, Y.E. and Ifo, S.A. 2017. Age, extent and carbon storage of the central Congo Basin peatland complex. Nature Publishing Group. [Online]. **542**(7639), pp.86–90. Available from: http://dx.doi.org/10.1038/nature21048.*

2, 22: I think it may be more accurate to say "Prior to this conversion, drainage canals were installed" since I think the draining was largely completed as part of the "mega rice project" before the massive emissions of the 82-83 El Nino. Interesting side note, the drainage canals were built at the recommendation of a USFS expert despite the same type of drainage causing an environmental smoke disaster in the US in the 1930s as described in "Sand County Almanac."

*We would like to thank the reviewer for providing some background information on the drainage canals. Our statement refers more broadly to the process of conversion of tropical peat forests to plantations rather than being specifically for the mega rice project.*

2, 27: surface fires are not typically extinguished in Indonesia, but just burn out.

*There are increasingly efforts to extinguish peatland fires by fire fighters. However we agree that it is uncertain how successful these attempts are, and have changed 'extinguished' to 'gone out'.*

3, 1-2: Here and throughout please be careful to use "peatland" to indicate all the vegetation and peat in areas underlain by peat or "peat" to indicate just "peat". These statements should be phrased more quantitatively similar to what is presented later in the text

*We would again like to thank the reviewer for pointing out confusing language. We have noted this and changed 'peat' and 'peatland' as appropriate to the correct meanings throughout the text*

3, 10: "water content" of peat may be replaced by something about the "water table", i.e. peat can burn down to the groundwater level

*We agree that the level of the water table is important to the burn depth, however in this study we have used soil moisture, which is closely related to water content. We have therefore included both 'water content' and 'water table':*

*'Burn depth depends on the level of the water table and the water content of the peat, with increased burn depth when the water table is lowered and the peat dries out (Rein et al., 2008; Ballhorn et al., 2009; Huang and Rein, 2015)'*

3, 17-19: The vast majority of field data on gases emitted by smoldering peat were measured by Stockwell et al 2016 not Hu et al. who review a small subset of the gases measured. The Roulston et al paper never actually recommended any EF, they just said the instantaneous EF change during the fire. The gravimetric measurement of EFPM by Jayarathne et al., 2018 is likely the best value for reasons described later in review. Stockwell et al 2014 gave the first detailed discussion of variation in peat fire EF (Table 3 and Fig. 7) and Stockwell et al 2015 provide the most comprehensive EF data for peat fires.

*We would like to thank the reviewer for summarising the EF studies used. We change the citation of Hu et al. to Stockwell et al.*

Stockwell, C. E., Yokelson, R. J., Kreidenweis, S. M., Robinson, A. L., DeMott, P. J., Sullivan, R. C., Reardon, J., Ryan, K. C., Griffith, D. W. T., and Stevens, L.: Trace gas emissions from combustion of peat, crop residue, domestic biofuels, grasses, and other fuels: configuration and Fourier transform infrared (FTIR) component of the fourth Fire Lab at Missoula Experiment (FLAME-4), Atmos. Chem. Phys., 14, 9727-9754, doi:10.5194/acp-14-9727-2014, 2014.
Stockwell, C. E., Veres, P. R., Williams, J., and Yokelson, R. J.: Characterization of biomass burning emissions from cooking fires, peat, crop residue, and other fuels with high-resolution proton-transfer-reaction time-of-flight mass spectrometry, Atmos. Chem. Phys., 15, 845-865, doi:10.5194/acp-15-845-2015, 2015.

The statement "Roulston et al. (2018) and Wooster et al. (2018) found that assumed EFs for tropical peat fires could be underestimated by a factor of three" should be more quantitative. One can assume peat fire EFs are too big, or too small, by as much as they want. Roulston only said that the EF can change during the fire, but did not suggest any procedure that was more valid than random sampling. Wooster et al had experimental issues and their "peat EF" had variable contributions from non-peat fuels in all but one location as clarified more later.

*We have changed this statement to the following in order to make it more quantitative as suggested.*

*'Roulston et al. (2018) and Wooster et al. (2018) found that EFs for tropical peat fires could be underestimated by a factor of three (PM2.5 EF from peat fires is assumed to be 9.1 g kg-1 in GFED4, compared to 24 g kg-1 suggested by Roulston et al. (2018) and 28 g kg-1 suggested by Wooster et al. (2018)) .'*

3, 20: "EFs were"

*This has been changed.*

3, 21-22: Kuwata et al would likely be "peatland" fires not "peat" fires since they monitored from a great distance. The idea of being able to measure a precise annual average EF implied

here is probably not realistic. It's hard to measure CO2 from fires at a distance due to mixing effects and CO2 is the largest contributor to the EF. It's interesting that 13 and 19 average to 16 which is close to the 17 that Jayarathne et al essentially got. The year to year differences could be due to a number of factors besides the year. The uncertainties seem too low at e.g. 19 +/- 2.

*'Peat' has been changed to 'peatland' when referring to Kuwata et al.*

*We would like to thank the reviewer for bringing up these concerns with the method used in Kuwata et al. We are not trying to suggest that the difference between the Kuwata et al. EFs is due to the different years, only that it had changed, and this adds to the variations found in EFs. The phrase 'EFs can also vary between years' has been removed from this sentence.*

3, 2-27: I don't think the observations were scaled so rewrite: "Previous studies have scaled particulate fire emissions from global fire emission datasets, or simulated fire-derived aerosol optical depth (AOD) or PM, by a factor 1.36 – 3.00 in order to match observations" as "Previous studies have scaled particulate fire emissions from global fire emissions inventories by a factor 1.36 – 3.00 in order to match observations of PM or simulated fire-derived aerosol optical depth (AOD)"

*We agree this sentence was unclear. We have changed to" Previous studies underestimate measured aerosol optical depth (AOD) or PM, and scale particulate fire emissions from global fire emission inventories, or simulated fire-derived aerosol by a factor 1.36 – 3.00 in order to match observations."*

3, 29: "datasets" could be many things (including the papers with high EF) so change "datasets" to "inventories" throughout.

*We have changed 'datasets' to 'inventories' throughout.*

3, 33: lowering the water table exposes more peat to the possibility of burning.

*We agree that a lower water table exposes more peat, and have changed this sentence to say 'Severe droughts lower the water table, exposing more peat and increasing the susceptibility of burning'*

3, 34: The two biggest El-Nino peat fire events were 82-83 and 97-98. The biggest El-Ninos create fire episodes that span two years.

*We have included 1982-1983 and 1997 - 1998 in the years listed.*

4, 2-4: rewrite to clarify that Jayarathne is peat only and the others are for peatland.

*We have rewritten as suggested.*

4, 4-6: It's very important to be precise about mortality before the atmospheric community loses credibility. Smoke is a contributing factor to premature mortality, which is not like a perfectly healthy person "getting shot."

"responsible for" should be "contributed to" and "excess deaths" should be "premature deaths" so phrase reads: "contributed to between 6,513 and 17,270 premature ..." and fix line 5 same way.

*We agree there is a need for careful language here. To make our points clear we have included premature to make 'excess premature deaths' and have changed responsible for to 'may have caused'.*

4, 10-14: change dataset to inventory in 4 places?

*We have changed this wording throughout the paper.*

4, 30: Does dynamics and processes include evaporation and secondary formation, deposition, etc?

*SOA formation is from Hodzic and Knote (2014) using a lumped surrogate VOC for biomass burning, co-emitted with CO which oxidises with OH and condenses into SOA (Hodzic and Jimenez 2011). For aerosols and trace gasses, sub-grid-scale washout is part of the Grell & Deveni (2002) convection scheme. For trace gases, there is grid-scale washout using Neu and Prather (2012).* The dry deposition is based on the resistances approach (Wesely 1989).

*The following has been added to the model description:*
*'This includes an SOA scheme based on Hodzic and Jimenez (2011). Primary organic aerosols are considered as non-volatile in the model.'*

5, 6: POA volatility? SOA scheme? Relative amount of modeled secondary to primary emissions?

*POA is treated as non-volatile the model. The SOA scheme is based on Hodzic and Jimenez (2011). It produces reasonable concentrations of SOA from biomass burning (Hodzic and Knote, 2014). This has been added to the model description (see comment above).*

*In this model setup we assume a lumped SOA precursor from biomass burning emitted at a rate of 0.04g per g of CO emitted.*

*We now add CO emissions and SOA formation from biomass burning emissions to Table 4. SOA formation from biomass burning is estimated to be 3.12 Tg in our revised emissions (FINNpeatSM). This is equivalent to 43% of the primary PM2.5 emissions.*

*We clarify in the abstract that emissions refer to primary emissions of PM2.5.*

5, 16: burned per pixel

*This is actually the area burned per fire hotspot detected. We have edited this sentence to make this clear.*

5, 25: EFs come from the peat itself? They are based on some measurement, probably the one sample of Indonesian peat burned in a lab at the time Akagi 2011 was compiled. We started working towards peat fire field measurements in 1994 against large odds. In 2000 we were able to legally import and burn one sample in our lab. In 2015 we finally completed field measurements.

*We would like to thank the reviewer for showing that the wording used here could be confusing. The EFs come from studies on Indonesian peat. This sentence has been edited to 'studies of Indonesian peat fires' to make this clear.*
*The EFs come from Christian et al. (2003), which is where Akagi et al. (2011) get their peatland EFs from.*

*Christian, T.J., Kleiss, B., Yokelson, R.J., Holzinger, R., Crutzen, P.J., Hao, W.M., Saharjo, B.H. and Ward, D.E. 2003. Comprehensive laboratory measurements of biomass-burning emissions: 1. Emissions from Indonesian, African, and other fuels. Journal of Geophysical Research-Atmospheres. 108(D23), p.4719, doi:10.1029/2003JD003704.*

*Akagi, S.K., Yokelson, R.J., Wiedinmyer, C., Alvarado, M.J., Reid, J.S., Karl, T., Crounse, J.D. and Wennberg, P.O. 2011. Emission factors for open and domestic biomass burning for use in atmospheric models. Atmospheric Chemistry and Physics. 11(9), pp.4039–4072.*

6, 8: The surface fuels likely burn on the day of detection, but, if the peat ignites, having it release emissions over the next few days at least would be more realistic, but probably too late to change. Maybe it should be noted that this could shift the emissions forward in time. Also if a fire creeps along and is visible for many days hopefully it is counted as a fire all those days and not tossed as a duplicate on some days. (Later noticed this was partially addressed perhaps belatedly?)

*The reviewer is correct that having a delayed release of peat emissions might be more realistic. We have considered this issue, along with other additional complexities which could improve the emissions. We decided that for this study, the increased complexity of was unnecessary, given the already good match with observations we are getting. We aim to explore this in future work.*

6, 10: delete "a"

*We have fixed this mistake as suggested.*

6, 11: I think hotspot detection is widely believed to be more sensitive than burned area detection so I would instead expect many hotspots did not have an associated burned area. Can any attempt be made to use LandSat or SAR burned area data, which may be more sensitive burned area products?

*Tansey et al. (2008) suggest that satellite coverage, cloud coverage, or fire size/duration could be the reasons for hotspots not being detected.*

*We have considered the issues that could be associated with using hotspot data, however since the new emissions are a FINN product, we want to keep the basic emissions code the same as for FINN.*

*Tansey, K., Beston, J., Hoscilo, A., Page, S.E. and Paredes Hernández, C.U. 2008. Relationship between MODIS fire hot spot count and burned area in a degraded tropical peat swamp forest in Central Kalimantan, Indonesia. Journal of Geophysical Research. [Online]. **113**(D23), p.D23112. Available from: http://doi.wiley.com/10.1029/2008JD010717.*

6, 14: Ok, our crew in Indonesia noted surface areas that burned where the peat did not ignite by the end of the dry season so peat area burned could be less than surface area burned.

*We are glad to hear that this has been seen in the field.*

6, 15-17: Stockwell et al., (2016) measured burn depth on numerous sites (see their supplement) and report a close average value of 34 +/- 12 cm. Might be good to add to Table 1 and reference string for completeness. Similarly, Konecny et al., (2016) reported a peat bulk density of 0:120+/- 0:005 g/cm that could also go in Table 1?

*We would like to thank the reviewer for suggesting extra values to include in our average. These have been included in the study, and the averages in Table 1 remains at 37cm for burn depth, and 0.11g/cm for density.*

6, 22-30 and Table 2: I have a number of comments on the EF in Table 2. I'll go in order by study.

*We thank the reviewer for the detailed comments on the different studies synthesised on Table 2. Whist we acknowledge that some studies may arguably be more comprehensive than others, we would prefer not to weight one study more heavily than another. We calculate the average across a range of studies, so that any particular issues with one study will not greatly impact our overall results.*

Christian et al., (2003): This was just one sample of Sumatran peat burned in lab in 2000. One might be tempted to weight this study less than the more extensive studies and especially the extensive 2015 field studies of Stockwell and Smith.

*The reviewer is correct about this study only using one sample of peat. However, this is also the only study that we have found done before the 2015 fires, and one of few which gives EFs for a number of species, and we therefore prefer to keep the average as it is.*

Wooster et al., (2018): In general, it's important to clarify peat or peatland throughout the paper as noted above. It seems like one should not use peatland EFs for peat because FINN has EF and fuel consumption for surface vegetation already so applying peatland EF to peat

will double-count? I have enough concerns about the EF in the Wooster study that it's a bit of a challenge to determine the best explanation.

*We agree with the reviewer that it is important to clarify peat or peatland. We have tried to make it clear in Table 2 which studies were peat only and which may also include vegetation. We have also included the following*

*'Some of the EFs used have been calculated from fires on peatland, which also contain vegetation burning. '*

*We agree that using an EF calculated from peatland burning could lead to inaccuracy. However, because emissions are dominated by peat combustion rather than combustion of vegetation this error is likely to be relatively small.*

1. When using light-scattering to measure PM, the response depends on particle density as they noted, but also the size distribution. They took a good step to calibrate the light-scattering with
filters also collected during the measurements. However, the graph shows many calibration points being collected at extremely high concentrations (e.g. 17-25 mg/m3 when .3-1 mg/m3 is already stretching normal measurement level). That means some PM data could be affected by unrepresentative gas to particle partitioning and greatly inflate some EFPM. This doesn't seem to have impacted their "pure peat results" from "location 5" (vide infra), but may be why they got EFPM for smoke that reflected some burning surface vegetation that were in their words "among the highest ever recorded" and well above the literature average for various surface vegetation types.

2. They use a significantly higher %C for peat than the other studies, which increases all EF across the board without saying how many peat samples were measured or how. It would have been nice to know if there was one or many peat samples (see comment on Stockwell 2015).

3. They left the carbon in the PM out of the carbon mass balance in their EF calculations. Along with ignoring NMOG this would inflate all EF by ~5%. In addition, in a unique, unexplained step; they report two different EFPM for every site depending on using CO or $CO_2$ as a reference.

4. They report EF for (1) "peatland" or "mean of all locations"; (2) "peat only" which their text says actually is "mostly peat" and includes some burning surface vegetation; (3) "really pure peat only" (location 5). For PM there are two sets of EFPM for each of these categories. In their table the EFPM range from 34.4 to 11.4. After reducing the EF for location 5 (actual pure peat) by 5% (to better reflect true total carbon emissions) the EFs are $CO_2$ (1627), CO (308), $CH_4$ (4.96), PM-avg (12.6). If any peat fire EF data were to be used from this study, then these seem appropriate and not the values in the table. In particular, I'm not sure where the 28 for EFPM the authors quote came from.

*We would like to thank the reviewer for detailing their concerns about this study. We agree that there may be some issues with their methods. By using an average across multiple studies, this reduces the impact of individual studies on our results.*

*The value of 28 g/kg is derived in the Wooster et al. paper:*
*"Using the data of Table 4, we derive a mean peatland EFPM2.5 of 28 ± 6 g·kg−1, calculated as with the carbonaceous gas EFs from a 73% weighting atmospheric chemistry transport models (CTMs)].".*

Stockwell et al 2016: Stockwell and Jayarathne co-deployed in Borneo. Stockwell reported a subset of the filter-based EF data to interpret simultaneously-collected PAX data. Thus the Stockwell OC and PM should not be included, but the BC can be since it is an independent measurement by PAX rather than thermal-optical analysis. This Stockwell study may be most extensive high quality field measurements of pure peat emissions for BC and trace gases.

*We thank the reviewer for this information. This is now indicated in the Table.*
*'Stockwell et al. (2016) and Jayarathne et al. (2018) were co-deployed on the same study'*

Stockwell et al., 2015: This was a lab study, not a field study. It's probably not included in the paper, but nine samples of Indonesian peat measured in this study ranged from 50 to 61% C and from 50 to 58 %C when nominally at the same site; illustrating the heterogeneity and the need for intensive sampling of this parameter. The value 0.57 +/- 0.025, based on the 7 most representative samples, was used in this study, Stockwell 2016, and Jayarathne 2018.

*We would like to thank the reviewer for pointing out this mistake. The table has been updated to include this as a lab study.*

Jayarathne et al., 2018: Gravimetric filter sampling has challenges, but is often considered the most accurate approach for PM, OC, EC, etc EFs. OC/PM ratios are best measured on the same sample.

Table 2 general comment. I think it would be best not to recycle any confusion regarding Jayarathne/Stockwell overlap and peatland versus peat. If that meant that the real best EF were different from what is shown as the Table 2 average and used in the model; and the model could not be run again, that would be understandable, but then the discussion could speculate that secondary PM or biomass burned was actually higher or add other topics.

*Thank you for these useful clarifications. We also thank the reviewer for understanding that any change in our calculated EF would require a complete rerun of our simulations which is not possible at this stage.*

*We have updated Table 2 to provide information on the issues and clarifications raised by the referee. We think the revised table will provide readers with the necessary information to understand the different data sources. We add a discussion to our paper to discuss these issues.*

*We agree that the PM2.5 EF used from Wooster et al. may result in over counting of vegetation emissions. However we think that the value suggested by the reviewer of 12.6 g/kg may also be unsuitable. Using an EF calculated from just one location introduces extra uncertainty. Wooster et al. also provide an average of the 'peat only' fires which gives a PM2.5 EF of 22.25 +/- 8.33 g/kg.*

*We have recalculated the emissions, removing Stockwell et al. (2016) PM2.5 and OC EFs, and using the 'peat only' PM2.5 EF from Wooster et al. (22.25 g/kg). This changes the following average EFs to:*
*Old EF (g/kg) → New EF(g/kg) (percentage of old EF)*
*PM2.5: 22.3 → 19.78 (89%)*
*Organic carbon: 11.5 → 9.21 (80%)*

*We have also recalculated the emission using these new values. The difference to total emissions of PM2.5 and OC for September and October are:*
*Old total → new total (percentage of old total)*
*PM2.5: 7.33 Tg → 6.98 Tg (95%)*
*OC: 3.84 Tg → 3.26 Tg (85%)*

*Calculating EFs in a slightly different way using a different set of studies only changes our calculated PM2.5 emissions by 5%. We therefore are confident that our calculated EFs are relatively robust to such assumptions.*

*The following has been included when describing the emissions to reflect to possible over counting of emissions.*

*'The PM2.5 EF used for FINNpeat and FINNpeatSM is at the higher end of the range of values used in other studies. The same PM2.5 emissions combined with a lower EF would require a greater burn depth or greater area burned per fire hotspot.'*

7, 5: Stockwell et al., (2016) showed brown carbon absorption is very important for peat smoke.

*This sentence is about black carbon having a minor contribution to the total mass. Stockwell et al. (2016) state that 'The emissions of BC [black carbon] were negligible'*

7, 11 – 9, 3: This is an excellent idea to vary burn depth with guidance based on soil moisture – though water table may be even better (Putra et al., 2018). Did the soil moisture in high fire regions get significantly lower than the near constant regional average reported?
Perhaps in a future version it can be added that burning that deep can take time and that there is a lag between rainfall and water table level.

Putra, E.I., M.A. Cochrane, Y. Vetrita, L. Graham and B.H. Saharjo. 2018. Determining critical groundwater level to prevent degraded peatland from severe peat fire. IOP Conference Series: Earth and Environmental Science 149 012027 doi: 10.1088/1755-1315/149/1/012027

*We would like to thank the reviewer for this comment. We used soil moisture rather than the water table level as satellite soil moisture is available. We hope that the soil moisture data gives a representation of the water table level. Figure S2 shows the average soil moisture for two high fire areas. We have also calculated a soil moisture average for two low-fire areas, and Figure S2 has been updated to show these areas. The soil moisture is lower in the high-fire areas (Orange and Grey) than low-fire areas (Green and Purple).*

*We agree that there may be a lag between the surface soil moisture and the water table (and therefore the burn depth), however, since we are not using the water table level, we are not sure what the lag might be. As a trial, we have tried recalculating the emissions with the burn depth based on the soil moisture from 5 days previously. The daily PM2.5 emission are shown in the figure below. It makes a small difference to the final total PM2.5 emissions, most noticeably to the peak emissions. Without more knowledge on the relationship between soil moisture and burn depth, however, we think it is better to not include a lag.*

[Figure]

Also, in early November at precisely-located sites in Kalimantan, Stockwell et al 2016 (supplement) report actual burn depths for site specific comparison; or one might compare to the average of 34 cm. However, their data may be biased to dry spots that were still burning? As reported in Stockwell reaching the maximum depth takes time and termination of the fires likely requires the water table to rise since peat is difficult to "rewet".

*Thanks for these comments. We agree that there are many complexities that will not be captured by our analysis.*

9, 5: Excellent feature to experiment with injection altitude.

*We would again like to thank the reviewer for this comment.*

9, 8: Smoldering peat can release smoke very close to the ground (pics in Stockwell 2016 supplement).

*We agree that smoke from smouldering fires may be released at the surface, this is why we have included the surface injection option. This sentence is referring to the fire emissions in general, rather than just the peat emissions. We have included the following, to reflect this comment:*
*'However, smoke from smouldering peat fires can be emitted close to the ground. '*

9, 15: These are two excellent choices for plume rise experimentation in model.

*We would like to thank the reviewer for their agreement with the options we have chosen.*

9, 22 and 11, 3, Figure 1, Table 3: There is more than one PM monitoring site in Indonesia. The government agency BMKG has seven PM10 sites (http://www.bmkg.go.id/kualitas-udara/informasi-partikulat-pm10.bmkg) and 5 PM2.5 sites: (http://www.bmkg.go.id/kualitas-udara/informasi-partikulat-pm25.bmkg). Include a sentence to mention this and why not used (no access?)? I have 17 days of PM10 from 2015 for Palangkaraya that I happened to save on my hard-drive and it may be possible to get the rest. Indonesian sites could have high value for constraining emissions by virtue of being less sensitive to secondary processes, meteorology, and other sources. They would have high S:N. I also have Oct 2015, Palangkaraya visibility, which may be useful.
Any estimate of how burning in peatlands in "West Papua" might impact model/ambient-PM agreement?

*We would like to thank the reviewer for these suggestions. We agree that measurement sites closer to the fires would be very useful for our study. We have contacted the reviewer separately about accessing data from these stations. Unfortunately, the data is not easily available, and we do not think that it will be possible to get for this study. We are striving to gain access to the BMKG data, which is not available freely to the public.*

*For this study we have focused on the fires in Kalimantan and Sumatra, and have not included West Papua in our domain, either for the model runs or for the emissions calculations. However, we suspect that, given the distance between the observations and West Papua, and the large fires that were occurring in Kalimantan and Sumatra, much closer to the stations, the fires in West Papua would only have a small contribution to the measurements.*

9, 31: PM1/PM2.5 seems high, usually about 80%, and that's what they saw on next page.

*We agree with the reviewer that this value seems high, however it is what the study found.*

11, 11-12: Why comparing to two months instead of whole season?

*The majority of the fires in 2015 were in September and October. Previous studies (Jayarathne et al. and Wooster at al.) looked at emissions from September and October, so we matched this in order to compare with their findings.*

11, 14: Should it be explained somewhere that Wooster et al based estimates on an inversion using MOPPIT CO (and give MOPPIT version since last version supposedly has improved sensitivity in boundary layer, but still a lot of uncertainty)?

*We have added a brief explanation of how the dry matter estimate was calculated in Wooster et al.*
*'Wooster et al. (2018) estimated 358±107 Tg of dry matter consumption for Kalimantan and Sumatra in September and October, using satellite CO emissions (from MOPITT) and a CO EF.'*

11, 15: "reasonable agreement"

*We have changed this to 'reasonable agreement' as suggested.*

11, 24: Jayarathne was estimating peat only. FINN + Jayarathne peat is 900 Tg right between FINNpeat and FINNpeatSM.

*We would like to thank the reviewer for pointing this out. We have made changes to reflect this. The sentence*
*'Jayarathne et al., (2018) estimated 547±259 Tg of $CO_2$ were emitted over South Sumatra and Kalimantan, suggesting that the total $CO_2$ emissions from FINN (353 Tg for Borneo and Sumatra) are too small, due to lack of peat fires in this dataset. '*

*Has been changed to:*
*'Jayarathne et al., (2018) estimated 547±259 Tg of $CO_2$ were emitted from peat fires over South Sumatra and Kalimantan, a range which includes the total $CO_2$ emissions from peat fires for FINN+GFEDpeat (469 Tg), FINNpeat (661 Tg) and FINNpeatSM (428 Tg).  '*

11, 31: "due mainly" - other things vary too, like burned area and EFPM for surface vegetation types? And should "peat" be "peatland"

*We have changed this to 'due mainly'.*
*This time, it should be 'peat', as this is reflecting the peat EF that we have used (which does include some values from peatland, as explained earlier, but we are still referring to as peat emissions).*

12, 4: Does "smaller area" indicate that the burned area of Whitburn et al 2016 was smaller than the burned area in FINN and GFED? In general, maybe the peat consumption and other data in Whitburn et al could be compared to the products developed in this study.

Whitburn, S., Van Damme, M., Clarisse, L., Turquety, S., Clerbaux, C., and Coheur, P. F.: Doubling of Annual Ammonia Emissions from the Peat Fires in Indonesia During the 2015 El Niño, Geophys. Res. Lett., 43, 11007–11014, https://doi.org/10.1002/2016GL070620, 2016.

*Smaller area refers to the study area in Jayaranthe et al. being smaller than our study area, so it is possible that the total emissions will be missing some of the fires that have been included in ours. This has been changed to 'for a smaller study area' to make this clear.*

*We have included the fuel consumption from the Whitburn et al. study as a comparison*

*'Whitburn et al. (2016), have estimated dry matter fuel consumption of 525 Tg, calculated using satellite CO emissions (from the IASI instrument) from peatlands, and the CO EF for peat from GFAS. This estimate is larger than that found for GFED or FINNpeatSM, and smaller than that found for FINNpeat.'*

12, 5: "emission for peat only was" – Jayarathne et al estimated only the peat and their peat only value for PM is right between FINNpeat and FINNpeatSM peat only PM emissions. Their uncertainty is large, but properly propagated and useful since no other uncertainties are reported:)

*This has been changed.*
*The sentence now reads:*
*' Jayarathne et al. (2018) found that, for a smaller study area in Sumatra and Kalimantan, the total $PM_{2.5}$ emission from peat fires was 6±5.5 Tg, a range which covers the total $PM_{2.5}$ emissions from peat fires from FINN+GFEDpeat (2.51 Tg), FINNpeat (8.51 Tg) and FINNpeatSM (5.24 Tg).'*

12, 12 or later in paper: There are some more studies to compare to and it should be clear what geographic area the peat/PM ratios represent in each study. Per his short comment, Eck estimated 80-85% of aerosol was from peat combustion at Palangkaraya where the peat contribution is probably higher than in Singapore.
Eck cites Kaiser et al who got 83% of emissions from peat and Wiggins et al who got 85% of emissions from peat.

Kaiser, J. W., van der Werf, G. R., & Heil, A. (2016). Global climate bi       omass  burning  in "State of the climate in 2015". Bulletin of the American Meteorological Society, 97(8), S1–S275. https://doi.org/10.1175/2016BAMSStateoftheClimate.1

Engling et al., 2014 got 76% peat fire impact on TSP at Singapore for hazy days in 2006.

Engling, G., He, J., Betha, R., and Balasubramanian, R.: Assessing the regional impact of Indonesian biomass burning emissions based on organic molecular tracers and chemical mass balance modeling, Atmos. Chem. Phys., 14, 8043-8054, https://doi.org/10.5194/acp-14-8043-2014, 2014.

*We would like to thank the reviewer for suggesting extra studies to include in our analysis. The finding by Eck et al., Wiggins et al. AND Engling et al. have been included in the results section, as a comparison to Figure 8:*

*'For 2015, Wiggins et al. (2018) suggest that ~85% of the smoke reaching Singapore was from peat fires, slightly higher than the contribution of peat fires to the simulated $PM_{2.5}$ concentration shown in Figure 8 (67%). Engling et al. (2014) found that in 2006, 76% of particulate matter in Singapore was from peat fires. At the Palangkaraya AERONET site in Kalimantan, Eck et al. (2019) found that 80-85% of AOD came from peat burning, consistent with the contribution of peat fires to PM2.5 simulated by the model at that location (Figure 8)'*

*Kaiser et al. (2016) show the peat fire contribution to total carbon emissions, which we have not calculated. We have therefore not included this study.*

Possibly useful?
Hansen, A. B., Witham, C. S., Chong, W. M., Kendall, E., Chew, B. N., Gan, C., Hort, M. C., and Lee, S.-Y.: Haze in Singapore – source attribution of biomass burning PM10 from Southeast Asia, Atmos. Chem. Phys., 19, 5363-5385, https://doi.org/10.5194/acp-19-5363-2019, 2019.

*This study shows the contribution of fire emissions in different regions to PM in Singapore at different times of the year. We have included it in the results section, with other studies which show similar:*

*'Hansen et al. (2018), found that during August to October fires in South Sumatra and Central Kalimantan are the largest contributors to $PM_{2.5}$ in Singapore.'*

13, 5-9: How much do emissions from the island of New Guinea impact the monitoring stations employed in this study (see Fig. 2 in Hansen et al above)? Also, line 9, "need for future"

*Figure 2 and Figure 8 in this study suggest that fires on New Guinea contribute very little to the PM in Singapore (<1%). This supports our thoughts (see previous comment of fires in West Papua) that, due to the distance between these fires and the monitoring stations, compared to the much closer fires in Kalimantan, these fires will not contribute much PM to the monitoring sites we have been using.*

13, 18: The word "matches" is used here and elsewhere when maybe "consistent with" is better or a % difference would be more useful.

*This has been changed to 'is consistent with', and matches has been changed in two other places:*
*20, 12: 'matches' → 'is similar to'*
*20, 16: 'matches' → 'is consistent with'*

13, 19-25: Would it help if the model released the peat emissions over the weeks following detection? Peak regional PM in Palangkaraya was in October rather than September.

*This is a good suggestion. We have considered the idea of having a delayed release of emissions after a fire, and, as a simple test, the total PM2.5 emissions for each month have been calculated if half the emissions are released on the day of the fire and half spread over a two week period after the fire. This results in 37% of the year's emissions being released in September and 36% in October (compared to 36% and 32% with no delay on the emissions). So implementing a delayed release of emissions could show an improvement. However, applying this delay to individual fires in the emissions is more complex than to the total emissions, and it was decided that the added complexity was unnecessary for this study, especially as there is little information on exactly how the emissions should be released over time.*

Figure 4: FINNpeat and FINNpeatSM seem to overestimate PM and increasingly so with distance in the two examples shown. One could argue that the sites closer to the fires and with higher PM are best for constraining initial emissions or EFs for PM. Downwind sites are more subject to secondary formation, evaporation, deposition, or transport errors. However, Fig. 5 does show the lowest bias overall and the highest correlation overall for FINNpeatSM, which is useful. Is there value in comparing the integrated amount of PM modeled vs measured over the season (or by month) by site in a Table? Is there any CO monitoring to compare to?

*We agree with the reviewer that downwind sites are more subject to post emission processes in the model. Although there seems to be an improvement in the comparison closer the fires in Figure 4, this is only two sites. This pattern is not seen across all sites, although this could be because we have little data from sites close to the fires. We think the integrated comparisons across sites are more valuable than analysis at individual locations. Access to data from stations in Indonesia is a priority for future work.*

*Unfortunately we do not have access to any CO observational data.*

14, 5-8: Surface fuels burn quickly after detection and then the peat smolders till the end of the dry season. At the end of the dry season we saw an average burn depth of 34 cm, but burning down to that depth and or laterally across much of the burn scar takes time. The model choice to shift all peat emissions forward to the day of detection could cause Sept/Oct too high? On lines 6 and 7, be specific using "model" or "models", all of them, or which ones?

*We have discussed the disadvantages of having the emissions released on the day the fire is detected.*

14, 13-16: The effect of a non-linear relationship might also be mimicked by a longer time-frame for peat consumption as speculated above.

*There are many different factors which could affect the emissions. The burn depth/soil moisture relationship is one of them.*

14, 16: Has a relationship between soil moisture and burn depth in fact been measured? The Putra et al reference cited above discusses a large data base relating rainfall, water table, and burn depth.

*We have not been able to find a study linking soil moisture and burn depth. Our work gives a first indication that soil moisture might give some useful information about depth of burn.*

14, 18-20: This seems to finally acknowledge the point about the emissions taking time.

*Our methods section (P6, L7) describes our method of emission. We modify this to*
*"For each fire, the corresponding emissions are released on the day that the fire was detected, with no long-term smouldering effects which may be important for peat fires."*

15, 1-2: "Only hotspots that last 3 days are a fire" makes no sense so it's good the authors ignored this. There could be cases where a 3-day fire is needed to ignite wetter peat.

*Syaufina and Sitanggang refers to the possibility that hotspots lasting less than three days are often not a fire. This should read 'that last at least 3 consecutive days' and has been changed to this.*

15, 3: The timing of the peaks in Pekanbaru is excellent.

*This is one reason why we feel that introducing a delayed release of emissions may be adding unnecessary complexity to the emissions. The model seems to do a good job of simulating the timings of the peaks, just not the magnitude, and changing the timings of the emissions may adversely affect this.*

16, 5: Without knowing anything about secondary processes the best way to validate EF is with EF measurements and the authors are using updated EF, but undetected burned area can't really be written off? There could be undetected fires that are offset because the model could e.g. underestimate evaporation?

*This could be true, we have rephrased the sentence to reflect this.*

*'While this is likely to effect the emissions, our work suggests that PM emissions in GFED4 could also be underestimated because EFs for peat combustion are too small. '*

17, 1-2: Injection throughout the boundary layer, as opposed to all at the surface always reduces the modeled PM. Per above, there could be missing fires that are compensated for in the injection choice. Is there any model-assumed evaporation associated with faster dilution into the whole boundary layer?

*We agree that there are a range of factors that may compensate but are difficult to verify without measurements of the vertical profile of aerosol over these fire impacted regions.*

*The treatment of aerosols is the same throughout all model layers, so there is no difference between the surface and boundary layer.*

17, 12: Is POA inert in model? A recent study suggests about 50% of PM can be lost by evaporation (or deposition) for some fires.

Selimovic, V., Yokelson, R. J., McMeeking, G. R., and Coefield, S.: In situ measurements of trace gases, PM, and aerosol optical properties during the 2017 NW US wildfire smoke event, Atmos. Chem. Phys., 19, 3905-3926, https://doi.org/10.5194/acp-19-3905-2019, 2019.

Other BB studies (e.g. Vakkari et al., 2014; 2018) saw SOA dominate.

*POA is treated as non-volatile in the model, so there is no evaporation. SOA is included.*

*We agree with the reviewer that evaporation and secondary formation of OA may be important, although there is still uncertainty around the impact of these processes. Jolleys et al. (2012) found that there was little SOA production from biomass burning, and only a small net loss of OA. Brito et al. (2014) found that there was little change to OA downwind of fires.*

*Brito, J., Rizzo, L. V., Morgan, W.T., Coe, H., Johnson, B., Haywood, J., Longo, K., Freitas, S., Andreae, M.O. and Artaxo, P. 2014. Ground-based aerosol characterization during the South American Biomass Burning Analysis (SAMBBA) field experiment. Atmospheric Chemistry and Physics. 14(22), pp.12069–12083.*

*Jolleys, M.D., Coe, H., McFiggans, G., Capes, G., Allan, J.D., Crosier, J., Williams, P.I., Allen, G., Bower, K.N., Jimenez, J.L., Russell, L.M., Grutter, M. and Baumgardner, D. 2012. Characterizing the aging of biomass burning organic aerosol by use of mixing ratios: A meta-analysis of four regions. Environmental Science and Technology. 46(24), pp.13093–13102.*

17, 23-24: The author's model gets a better r value with AOD, not sure if this sentence is needed?

*Although this study finds that the model is good at simulating AOD, we felt it good to include a brief comparison of what other studies have found.*

17, 29: "mean" should be "peak"? Otherwise a mean of 1800 seems to conflict with the much lower means given just below?

*This is referring to an average over September and October, but for individual grid cells, i.e. the highest values seen in the red sections in Figure 7. The values below are referring to an average of September and October and over the domain. This has been changed to*

*'Simulated PM$_{2.5}$ concentrations from fires are greatest over Sumatra and southern Kalimantan, with simulated September-October mean concentrations exceeding 1800 µg m$^{-3}$ in some grid cells with FINNpeatSM emissions'*

18, 1-6: Are these peat/PM values for whole model domain or some subset? The values are higher than for the study of Lee et al discussed on next page

*These are for the whole study area for lines 1-2, and for Sumatra and Borneo for line 5-7. These values are referring to the % of PM2.5 concentration which comes from peat fires, whereas the values for Lee et al. are referring to the % of PM2.5 form fires in Sumatra.*

18, 6: So does this discussion then indicate that PM/CO increases slightly with aging. E.g. a factor of 76/71 still fairly close to the fires? Also, is this really known to be all from SOA when there are also inorganic precursors like NH3, NOx, SO2, etc?

*It is difficult to make any firm statements from these comparisons. We have rephrased this sentence with 'likely', to show that this is a suggestion.*

18, 11-12: don't understand this.

*Including peat fire emissions gives the largest increase in emissions where the large areas of peat are, i.e. Sumatra and Wester Kalimantan, rather than increasing emissions everywhere. This effects where the PM2.5 concentrations are increased.*

19, 3: Does current study agree that half or more of the pollution during haze episodes in Singapore is not from fires?

*The point here is not that 50% of the PM2.5 in Singapore is from fires, but that it is from fires in Sumatra. So more than half of the pollution is from fires.*

19, 4-6: not sure what this adds?

*Just some more context on how the location of the fires can be important. We have added the following sentence to this section to explain.*
*'The location of fires can be an important factor of their contribution to air pollution in populated areas.'*

19, 8-20, 2: More sites close to the fires (like Palangkaraya) could help determine the proper choice of injection altitude? Maybe it makes sense to inject non-peat emissions throughout the BL and peat fire emissions at the surface?

*We agree that more observations from close to the fires would help to determine the better injection method. However, without these observations, we can say that the injection options make limited difference except close to the fires.*

*Injecting non-peat emissions into the boundary layer and peat emissions at the surface could work, but no tests have been done to suggest that this would be an improvement.*

20, 7: In the AOD calculation, is the air-mass factor impacted at some PM level, is BrC ignored at 550, is it possible to compare to inferred AOD at Palangkaraya in Eck et al?

*We do not understand what the referee means by "is the air-mass factor impacted at some PM level". BrC is not included in the model, so is not included in the AOD.*

*The following has been added to the paper to show that the high AOD values from the model are similar to those estimated by Eck et al.*

*'The AOD simulated by the model exceeded 10 during September and October similar to the values estimated by Eck et al (2019) for the same period.'*

20, 11: add "surface" before "PM2.5" to make this conclusion even more obvious?

*We have added 'surface before 'PM2.5' as suggested'.*

20, 16: Here and earlier, a more inclusive geographic term than "Indonesia" might be "insular Southeast Asia."

*We have included Malaysian Borneo to make it clear where the fires we are referring to are.*

21,1: "datasets" to "inventories"? If not specifying inventories, should the qualifier "some" be included since there is agreement with some previous work?

*We have changed datasets to inventories throughout the text, and included 'some' for this sentence.*

21, 3: This study simulated more than just Sept-Oct so why limit to those months?

*We have changed this to be 'August –October'*

21, 8: fix "a new emissions datasets".
The new EF were likely applied to peat fires in Malaysia too? There can be many peat fires in Sarawak (one part of Malaysian Borneo). The Smith et al study measured peat fire EF on mainland Malaysia. Also lines 26, 28, 32 may overly single out Indonesia, which dominates but doesn't account for all the peat combustion in the model domain.

*We have fixed this to 'a new emissions inventory'*
*For this study we have only applied our new peat emissions to fires in Indonesia.*

21,12: mention agreement with some other estimates?

*We have included the following sentence:*
*'This total emissions agree with estimations by Wooster et al. (2018) (9.1±3.2) and Jayaranthe et al. (2018) (6±5.5 Tg from peat fires).* '

21, 13: improved estimates of peat consumption should improve CO2 as well?

*While the estimates of peat consumption are an improvement on FINN emissions (which previously had no peat consumption), it is hard to say if they are an improvement on the GFED peat consumption. The main differences between our peat emissions and GFED emissions are the EFs, which for CO2 are very similar to GFED.*

21, 19: Need work on secondary processes also.

*We have included a sentence on this:*
*'Further work is also needed to assess the impacts of secondary processes within the model on PM2.5 concentrations, and how this may affect the comparisons between model and observations made in this study.'*

21, 24: Work is needed on how rainfall, soil moisture, water table depth, burn depth, and burn rate or timing are related. In Putra et al TRMM rainfall had a minimum in August, but measured ground water level in an extensive series of dip wells hit a minimum one or two months later.

*We have included a sentence reflecting this:*
*'There is little data available on the relationship between surface soil moisture and burn depth, more work on this could lead to further improvement in the simulation.* '

21, 31: "impact" better as "contribution" since the ambient PM itself was not revised

*We have changed this.*

**Review of "Revised estimate of particulate emissions from Indonesian peat fires in 2015," by**
**Kiely et al.**
This paper revisits estimates of smoke emissions from the Indonesian fires of 2015. As is well known, fire emission inventories are highly uncertain, with large discrepancies among them. Here the authors take the FINN emissions as a base case, and then attempt to improve these emissions.

Fire emissions from peatland is expected to be a large contributor to smoke in this region, but the standard FINN inventory does not consider emissions from peatland. In addition, the depth of peat burned, a quantity governed by soil moisture, may influence the magnitude of smoke emissions in this region. To address these weaknesses in the FINN inventory, the authors do the following sensitivity studies: (1) adding the peat emissions from the GFED inventory to FINN, (2) recreating peat emissions with land cover maps, and (3) using remotely sensed soil moisture to constrain the magnitude of peat burned. While the base FINN

simulation underestimates observations, all three efforts listed above improve the model match with observed surface PM and with aerosol optical depths (AOD). Sensitivity study #3 comes the closest to observations. The authors estimate that vegetation and peat fires increased PM2.5 concentrations over Sumatra and Borneo during September and October 2015 by > 125 μg m-3. Such a heavy particulate burden has large
significance for human and ecosystem health.

The paper is excellent, in my view, and should be published after the following comments are addressed.

*We would like to that the reviewer for taking the time to review this paper, and for all the comments given.*

**Main comment.**
On page 16, the authors claim that PM emissions in GFED4 are underestimated because emission factors for peat combustion are too small. In my view, the authors cannot so easily dismiss the large role of clouds and haze in these or any smoke underestimates. GFED4 relies in part on area burned in constructing emission inventories, as well as on active fire information. FINN relies only on active fire information. Both kinds of data can be obscured by clouds and haze. (See for example Kaiser et al., 2012.) The authors should be aware that the interference from clouds and haze can vary strongly from month to month and from fire to fire, depending on meteorology and the magnitude of the fire itself. Cusworth et al. (2018) found that emissions from agricultural fires in India were more strongly underestimated during high fire years. At least two fire emission inventories, GFAS and QFED, make cloud-gap adjustments to account for fires obscured by clouds/haze.

*We agree that cloud cover and haze may affect the detection of fires. However, we think that this study suggests it is more likely due to the emissions factors, since FINN+GFEDpeat and FINNpeatSM are based on the same fire detection (FINN), and FINNpeatSM shows an improved comparison. We have edited the following sentence to avoid disregarding the effect of cloud cover.*

*'In contrast, our work suggests that PM emissions in GFED4 are underestimated because EFs for peat combustion are too small.     '*

*to*

*'While this is likely to effect the emissions, our work suggests that PM emissions in GFED4 could also be underestimated because EFs for peat combustion are too small.     '*

Indeed the variation of the cloud/haze interference could explain the inability of simulations in this work to capture the lack of temporal trend from September to October in the observations. The observations show relatively similar values of surface PM2.5 and AOD in both months (Figures 5b and 6b), while all three sensitivity simulations show as much as double the value in September as in October. Applying GFAS emissions to their model, Koplitz et al. (2015) captured the observed lack of trend from September to October in monthly mean PM2.5 values over Singapore.

*We have also included a sentence about cloud cover in the results section as a possible reason for the mismatch between the observations and model.*

*'Alternately, an underestimation in October could be due to clouds, or haze caused by previous fires, blocking the detection of fires by the satellite.'*

**Minor comments.**
Page 5. The text says that the model meteorology is allowed to run freely through the month, and then is reinitialized at the beginning of the month. Is that right? Would this approach lead to a discontinuity in meteorology at monthly intervals? Maybe a reference would be helpful.

*This is correct, the meteorology has been reinitialised at the start of separate model runs, which for this study is at the start of each month. This is to ensure that the meteorology does not drift too much over the three month period.*

Page 5. In the FINN+GFEDpeat simulation, is there concern that the inventory double-counts emissions? In this inventory, the GFED peat emissions are added to the FINN vegetation emissions.

*The inventory should not be double counting, because the GFED peat emissions are only for the peat fires, and the FINN emissions are only for vegetation fires. So for a fire on peatland, there will be the peat emissions (GFED) + vegetation emissions (FINN), which gives the overall peatland emissions.*

Page 6. Citing Tansey et al. (2008), the authors state that "60% of burned areas did not have an identified hotspot, implying an area burned per MODIS hotspot of approximately 40 ha." Are these fires in Indonesia only?

*Tansey et al. (2008) looks at fires on a peat swamp in Kalimantan, Indonesia. This has been specified in the paper.*

Page 9. The text states, "Fires can inject emissions above the surface." The author should clarify that they mean the surface layer of air, which is xx meters in the model.

*We have added in the following to make this clear*

*'Fires can inject emissions into air above the surface layer of the model, which in this model set-up is about 70 m.'*

Page 11. What exactly comprises PM2.5 in the model? Is this mainly BC and OC?

*PM2.5 in the model is comprised of SO4, NO3, Cl, CO3, NH4, Na, Ca, OIN, OC, BC, and water. The largest component is OC.*

Page 12. The authors should consider including emissions of BC and OC in Table 4.

*We decided not to include emissions of BC and OC as we have no comparisons with either observations or other studies for these components. We therefore restrict to emissions of PM.*

Page 13, Figure 2. The plots would be easier to interpret if low values were colored white.

*We have changed the colour scheme.*

Page 14, Figure 3. The caption should state what region is shown.

*The caption states that it is the area shown in Figure 1. We have also included the latitude and longitudes of this area. '95-120°E and 10°S-10°N'*

Page 15, Figure 4. The caption should provide the temporal correlation of modeled values and measurements for these two sites.

*We have included the r correlation for these sites in the caption.*

*'The r correlation for (a) is 0.47, 0.73, 0.52 and 0.50, and for (b) is 0.63, 0.60, 0.65 and 0.73 for FINN, FINN+GFEDpeat, FINNpeat and FINNpeatSM respectively.'*

Page 15. The text states, "The overestimation in September could also be due to an issue with fire detection. Syaufina and Sitanggang (2018) found that only hotspots which last for 3 days indicate fires, something which is not considered when calculating the emissions." This doesn't seem right. Agricultural fires in particular may last just hours. Indeed, a problem with active fire detection is
that the satellite overpass time may miss a short-lived fire, giving rise to the phenomenon that the authors note on page 6, with some burned areas not associated with any hotspot. In any event, if a hotspot lasts less than 3 days and is not a fire, what else could it be?

*Syaufina and Sitanggang (2018) does not specify what else may be causing hotspots if not fire, although, in correspondence with the authors, metal roofs on buildings was suggested. We have not included the idea of hotspots lasting several consecutive days in out emissions estimates, but feel that it is worth including as a possible reason for uncertainty.*

Page 18, Figure 6b. Wrong unit is given.

*We would like to thank the reviewer for pointing this out, it has been changed.*

Page 19, Figure 7. Again the authors should consider using a white color for very low values so that other colors stand out.

*We have changed this plot to the same colour scheme as Figure 2.*

Page 19, Figure 8. Colors in color bar are hard to see. Bar should be fatter.

*We agree that this is hard to see and have enlarged the colour bar.*

Page 20, Line 11. "Effected" should be "affected."

*This has been changed.*

Pages 20-21, Conclusions. The authors should consider briefly discussing the merits of other emission inventories used to simulate agricultural fires in Indonesia (e.g., GFAS).

*We have included the following sentence referring to GFAS emissions:*

*'Another commonly used emissions inventory is the Global Fire Assimilation System (GFAS1). This uses satellite fire radiative power to detect fires, combined with EFs to calculate daily emissions (Kaiser et al., 2012). For peat fires, some EFs are from studies of Indonesian peat, although the PM2.5 EF (9.1 g/kg) is from tropical vegetation, as is used in GFED. Reddington et al. found that GFAS1 requires the same scaling as GFEDv3 to match observations in Indonesia. It is therefore likely that GFAS1 would show similar results to GFED in our assessment. '*

Supplement page 3, Figure S3. Caption should state region shown. Also some statistics should be provided regarding the match between model and observations.

*The caption for Figure S3 has been changed to*
*'S3: 24 hour mean PM2.5 from observations in Singapore and model simulations with different fire emissions datasets and injection options. Solid lines are simulations with surface injections, dashed lines and simulations with boundary layer injection. 1:1 relationship shown by black dotted line. The fractional bias and r correlation of each comparison are given next to the plot.'*

*This plot has been updated, as the original plot was actually showing comparisons with only the North NEA station, rather than an average of all the stations as described.*

Supplement page 4-5, Figure S5-S6. Comment same as for Figure 7.

*The colour schemes have been changed.*

**References.**
Cusworth, D.H., et al., Quantifying the influence of agricultural fires in northwest India on urban air pollution in Delhi, India, Env. Res. Let., 13, 2018.

Kaiser, J.W., et al., Biomass burning emissions estimated with a global fire assimilation system based on observed fire radiative power, Biogeosciences, 9, 527–554, 2012.

Koplitz, S. N., et al., Public health impacts of the severe haze in Equatorial Asia in September-October 2015: Demonstration of a new framework for informing fire management strategies to reduce downwind smoke exposure, Environ. Res. Lett., 11, 094023, 2016.

**Short comment from Thomas Eck:**

A paper published just last week in JGR appears to be relevant to your study.
Eck, T. F., Holben, B. N., Giles, D. M., Slutsker, I., Sinyuk, A., Schafer, J. S., et al. (2019). AERONET remotely sensed measurements and retrievals of biomass burning aerosol optical properties during the 2015 Indonesian burning season. Journal of Geophysical Research: Atmospheres, 124. https://doi.org/10.1029/2018JD030182
https://agupubs.onlinelibrary.wiley.com/doi/10.1029/2018JD030182

*We would like to thank Thomas for suggesting this paper. They have looked at AOD values from AERONET stations during the Indonesian 2015 fire episode, and the findings agree with what we have shown in the paper.*

In particular, very high mid-visible AOD levels (∼11 to 13 at 550 nm) were estimated from extrapolation of near-infrared measurements. These values are very similar to those of some of your model estimates as shown in your ACPD Figure 6.

*The values estimated in this study are similar to those simulated by our model. We have included the following about this:*
*'The AOD simulated by the model exceeded 10 during September and October similar to the values estimated by Eck et al (2019) for the same period.'*

Additionally the estimated percentage contribution of peat to AOD in Palangkaraya, Indonesia was found to be ∼80% to 85% from AERONET aerosol absorption retrievals in our study, very similar to the values you show for this region in your ACPD paper Figure 8 for PM2.5.

*We have included the following sentence to discuss this:*
*'At the Palangkaraya AERONET site in Kalimantan, Eck et al. (2019) found that 80-85% of AOD came from peat burning, consistent with the contribution of peat fires to $PM_{2.5}$ simulated by the model at that location (Figure 8).'*

**New estimate of particulate emissions from Indonesian peat fires in 2015**

[revised manuscript text omitted]

20 HCN, $NH_3$ and PM (Stockwell et al., 2016). Until recently there have been few specific measurements of EFs for tropical peat fires. Roulston et al. (2018) and Wooster et al. (2018) found that  EFs for tropical peat fires could be underestimated by a factor of three ($PM_{2.5}$ EF from peat fires is assumed to be 9.1 g $kg^{-1}$ in GFED4, compared to 24 g $kg^{-1}$ suggested by Roulston et al. (2018) and 28 g $kg^{-1}$ suggested by Wooster et al. (2018)). There are large variations in EFs for peat in Indonesia. In one study measuring emissions from peat fires in Central Kalimantan during 7 days in 2015, $PM_{2.5}$ EFs were found to

25 vary between 6 and 30 g $kg^{-1}$ (Jayarathne et al., 2018).  Kuwata et al. (2018) used measurements from Indonesian peatland fires to estimate EFs of $PM_{10}$ of 13±2 g $kg^{-1}$ in 2013 and 19±2 g $kg^{-1}$ in 2014.

These uncertainties cause corresponding uncertainty in estimates of emissions from peat fires, and impacts on the regional air

30 pollution. Previous studies underestimate measured and aerosol optical depth (AOD) or PM, and scale particulate fire e  (Marlier et al., 2012; Ward et al., 2012; Johnston et al., 2012;

Tosca et al., 2013; Reddington et al., 2016; Koplitz et al., 2016). This suggests that particulate emissions from tropical peatland regions are underestimated in current fire emission inventories.

Severe fire events in Indonesia occur during periods of drought (van der Werf et al., 2008; Tosca et al., 2011; Gaveau et al., 2014; Field et al., 2016), resulting in strong seasonal and interannual variability. Severe droughts lower the water table,  more peat and increasing the susceptibility of burning.  Extensive fires and regional haze episodes across Indonesia have occurred in 1982-1983, 1997-1998, 2006, 2009, 2013 and 2015. During September to October 2015, dry conditions caused by a strong El Niño, resulted in large fires across Sumatra and Kalimantan. This fire episode was the largest in Indonesia since 1997 (Huijnen et al., 2016), releasing an estimated  188±67 TgC (Huijnen et al., 2016) as $CO_2$, and 149±71 TgC from peat fires (Jayarathne et al., 2018). The fires also emitted substantial amounts of $PM_{2.5}$ estimated at  9.1±3.2 Tg (Wooster et al., 2018), with 6.5±5.5 Tg from peat fires (Jayarathne et al., 2018). Particulate air pollution from these fires may have caused  between 6,513 and 17,270 excess premature deaths through short term exposure to fire-sourced $PM_{2.5}$ (Crippa et al., 2016) and as many as 100,300 excess premature deaths over the longer term due to exposure to this pollution (Koplitz et al., 2016).

[revised manuscript text omitted]

25  bias (defined in supplementary information) and correlation coefficients were used to evaluate the simulations. We did not use AOD data from MODIS retrievals, which significantly underestimated AOD over the region during this period, due to excluding smoke plumes that were mistook for clouds (Shi et al., 2019).

**Table 3: Observational data for 2015.**

| Data | Location | Time Period | Frequency of observations | Method | Reference / Source |
|------|----------|-------------|--------------------------|--------|--------------------|
| $PM_{2.5}$ | 5 Sites in Singapore: N: 1.41°N, 103.80°E E:1.33°N, 103.69°E C:1.34°N, 103.82°E W:1.33°N, 103.93°E S: 1.27°N, 103.83°E | 28 Sep 2015– 15[th] Nov 2015 | 1 hr | Thermo Scientific™ 5030 SHARP Monitor | National Environment Agency for Singapore |
| Composition resolved non-refractory $PM_1$ | National Technological University in Singapore 1.35°N, 103.68°E | 10 Oct 2015 – 31 Oct 2015 | 2-3 mins | Aerosol Chemical Speciation Monitor (ACSM) | Budisulistiorini et al. (2018) |
| $PM_{10}$ | Pekanbaru, Indonesia 0.52°N, 101.43°E | 1 Jan 2010 – 31 Dec 2015 | 30 mins | Measured using a Met One BAM 1020, Real-Time Portable Beta Attenuation Mass Monitor (BAM-1020) | |
| $PM_{10}$ | 52 locations across Malaysia | Aug-Nov 2015 | 1 hr | Measured using a Met One BAM 1020, Real-Time Portable Beta Attenuation Mass Monitor (BAM-1020) | Mead et al. (2018) |
| AOD | 8 AERONET sites | Aug - Oct 2015 | 24 hr average | Ground based remote sensing Sun photometer instrument measures intensity of solar radiation at 500nm wavelength, from which AOD is derived | AERONET version 2 |

**3 Results**

**3.1 Fire emissions**

Table 4 shows total dry matter consumption, $PM_{2.5}$,  $CO_2$, CO and SOA emissions from fires across Sumatra and Borneo

5   in September and October 2015. The dry fuel consumption is lowest for FINN (230 Tg), which does not include peat fires. Dry matter consumption is similar for GFED, FINN+GFEDpeat and FINNpeatSM (455 Tg, 514 Tg, 465 Tg respectively), and is highest for FINNpeat (612 Tg). This is likely due to the peat burn depth being greatest for FINNpeat. Wooster et al. (2018) estimated 358±107 Tg of dry matter consumption for Kalimantan and Sumatra in September and October, using satellite CO emissions (from the MOPITT instrument) and a CO EF. This dry matter estimate is in reasonable agreement with GFED and

10  FINNpeatSM. FINN estimates a smaller dry matter consumption compared to Wooster et al. (2018), whereas FINNpeat

**Table4: Total dry matter fuel consumption, $PM_{2.5}$,  $CO_2$, CO and SOA fire emissions for September and October 2015. Totals are shown for the area shown in Fig. 1.The percentage contribution from peat fires is indicated.**

|  | FINN | GFED | FINN+GFEDpeat | FINNpeat | FINNpeatSM |
|---|---|---|---|---|---|
| Peat fires included | No | Yes | Yes | Yes | Yes |
| Dry matter fuel consumption (Tg) | 230 | 455 | 514 | 612 | 465 |
| $CO_2$ emissions (Tg) | 353 | 773 | 822 | 1014 | 781 |
| Contribution from peat fires | 0% | 63% | 57% | 65% | 55% |
| $PM_{2.5}$ emissions (Tg) | 2.09 | 4.14 | 4.60 | 10.60 | 7.33 |
| Contribution from peat fires | 0% | 62% | 55% | 80% | 71% |
| CO emissions (Tg) | 20 | 75 | 77 | 109 | 78 |
| Contribution from peat fires | 0% | 80% | 74% | 82% | 74% |
| SOA from biomass burning (Tg) | 0.80 | 3.00 | 3.08 | 4.36 | 3.12 |

estimates greater dry matter consumption. (Whitburn et al., (2016), have estimated dry matter fuel consumption of 525 Tg, calculated using satellite CO emissions (from the IASI instrument) from peatlands, and the CO EF for peat from GFAS. This estimate is larger than that found for GFED or FINNpeatSM, and smaller than that found for FINNpeat.

5   Total September and October 2015 emissions of $CO_2$ follow a similar pattern to dry matter consumption, with similar values for GFED, FINN+GFEDpeat and FINNpeatSM (773 Tg, 822 Tg and 781 Tg), largest emissions for FINNpeat (1014 Tg) and smallest emissions for FINN (353 Tg). $CO_2$ EFs, are similar for GFED and FINN peat (1703 g kg $^{-1}$ and 1669 g kg $^{-1}$), explaining the similarity between dry matter consumption and $CO_2$ emissions for these inventories. The total $CO_2$ emissions for September to October estimated by Wooster et al., (2018) was 692±213 Tg, matching GFED, FINN+GFEDpeat

10   and FINNpeatSM. Jayarathne et al., (2018) estimated 547±259 Tg of $CO_2$ were emitted from peat fires over South Sumatra and Kalimantan, a range which includes the total $CO_2$ emissions from peat fires for FINN+GFEDpeat (469 Tg), FINNpeat (661 Tg) and FINNpeatSM (428 Tg).

15   Total emissions of $PM_{2.5}$ vary across simulations due to differences in assumed $PM_{2.5}$ EFs. FINN has the smallest total $PM_{2.5}$ emissions for September to October (2.09 Tg; Table 4). GFED and FINN+GFEDpeat have similar total $PM_{2.5}$ emissions (4.14 Tg and 4.60 Tg), smaller than that for FINNpeatSM (7.33 Tg) despite these inventories having similar dry matter consumption and $CO_2$ emissions. This is due mainly to the difference in the assumed EFs for $PM_{2.5}$ from peat fires, with 9.1 g kg $^{-1}$ used in GFED,

20   and 22.26 g kg $^{-1}$ used in FINNpeatSM. Wooster et al., (2018), assumed a $PM_{2.5}$ EF of 28±6 g kg $^{-1}$ and estimated that 9.1±3.2 Tg of $PM_{2.5}$ was emitted over the whole of Sumatra and Kalimantan for September and October 2015, similar to that found in FINNpeat (10.60 Tg) and FINNpeatSM (7.33 Tg). In contrast, FINN and FINN+GFED, which use the lower EF, produce smaller $PM_{2.5}$ emissions by a factor of 2 and 4 respectively. Jayarathne et al. (2018) found that, for a smaller study area in Sumatra and Kalimantan, the total $PM_{2.5}$ emission from peat fires was 6±5.5 Tg, a range which covers the total $PM_{2.5}$ emissions

25   from peat fires from FINN+GFEDpeat (2.51 Tg), FINNpeat (8.51 Tg) and FINNpeatSM (5.24 Tg).

GFED, FINN+GFED and FINNpeatSM all emit similar total amounts of CO over September and October (75 Tg, 77 Tg and 78 Tg respectively). This is likely due to the similar EFs used for peat fires (243 g kg $^{-1}$ in FINNpeatSM, 210 g kg $^{-1}$ in GFED). The total CO emissions from FINN are smaller (20 Tg) and from FINNpeat are larger (109 Tg). The contribution from

30   peat fires to total CO emissions (74 - 82%) is larger than for $CO_2$ (55 – 65%) and $PM_{2.5}$ (55% - 80%). For every 1g of CO emitted from fires, 0.04g of SOA is assumed, and the total SOA from each fire emission inventory is shown in Table 4. The SOA increases the total PM emitted from fires to 2.89 Tg from FINN, 7.14 Tg from GFED, 7.68 Tg from FINN+GFED, 14.96 Tg from FINNpeat and 10.45 Tg from FINNpeatSM.

Table 4 also shows the fraction of  emissions that are estimated to come from peat fires. Across the inventories that include peat burning, peat fires contribute 51-62% of dry matter consumption, 55-65% of $CO_2$ emissions and 55-80% of $PM_{2.5}$ emissions. The emission inventories with updated $PM_{2.5}$ emission factors result in a greater contribution from peat burning (71%-80%) compared to emission inventories with the older EFs (55%-62%). Wooster et al. (2018) found that peat fires contributed 85% of the dry matter fuel consumption, and 95% of the $PM_{2.5}$ emissions in September and October 2015,  greater than our estimates with updated EFs.

Figure 2 shows the spatial variations of the total $PM_{2.5}$ emissions during September and October 2015. In all inventories, greatest emissions occur in southern Kalimantan and central and southern Sumatra, matching the locations of peatlands (Fig. 1). For the FINNpeatSM emissions, Sumatra contributes 42% of the total $PM_{2.5}$ emissions, for FINNpeat, FINN+GFEDpeat and FINN, the contribution is 39%, 40% and 32% respectively. Wooster et al. (2018) found that 33% of the total $PM_{2.5}$ emissions came from Sumatra, while Koplitz et al. (2016) found that that 47% of OC and BC emitted in June to October 2015 came from Sumatra. Our estimates exclude fire emissions from eastern Indonesia. Nechita-banda et al. (2018) estimated that fires in eastern Indonesia and Papua New Guinea contributed around 15-20% of total CO emissions from fires across the region, highlighting the need future work to quantify PM emissions in this region.

FINN and GFED underestimate total emitted $PM_{2.5}$ and FINN underestimates total emitted $CO_2$ compared to the emissions found by Wooster et al. (2018) and Jayarathne et al. (2018), suggesting that peat fires are important contributors to these emissions. FINNpeatSM is the only emissions inventory that is consistent with  these previous studies for both $PM_{2.5}$ and $CO_2$.

Figure 3 shows daily total $PM_{2.5}$ emissions from the different inventories over the study area. Temporally, the inventories follow a similar pattern, with 80-90% of the total $PM_{2.5}$ emissions for 2015 occurring in August-October. For all the emissions inventories the majority of emissions are in September, followed by October and then August. GFED has the largest difference between September and October emissions (58% in Sep and 17% in Oct), followed by FINN+GFEDpeat (47% and 24%), FINNpeat (36% and 30%), and finally FINN (33% and 29%) and

[Figure]

**Figure 2: Total PM₂.₅ fire emissions during September-October 2015 (g /m2).**

[Figure]

**Figure 3: Total daily PM₂.₅ emissions from fires during 2015. Total shown for the area in Figure 1, 95-120°E and 10°S-10°N.**

FINNpeatSM (36% and 32%) which have the smallest differences between the two months. The reduced ratio of the fraction of emissions in September compared to October for FINNpeatSM is due to greater soil moisture in September resulting in a reduced peat burn depth.

Another commonly used emissions inventory is the Global Fire Assimilation System (GFAS1). This uses satellite fire radiative power to detect fires, combined with EFs to calculate daily emissions (Kaiser et al., 2012). For peat fires, some EFs are from studies of Indonesian peat, although the PM2.5 EF (9.1 g/kg) is from tropical vegetation, as is used in GFED. Reddington et al. found that GFAS1 requires the same scaling as GFEDv3 to match observations in Indonesia. It is therefore likely that GFAS1 would show similar results to GFED in our assessment.

**3.2 Comparison of model and observational data**

We evaluated the WRF-chem simulations with the different emissions inventories and injection options against measured PM₁₀, PM₂.₅ and PM₁ concentrations.

Figures 4 and 5 shows the comparison of simulated and observed PM concentrations. Comparisons of PM₂.₅ and PM₁ measurements only, which were restricted to Singapore, are shown in Fig. S3.

PM concentrations are underestimated by the model with FINN emissions, with a fractional bias (FB) of -0.67 with surface injection and -0.77 with boundary layer injection of emissions, with an average across both simulations of -0.72. The model with FINN+GFEDpeat emissions also underestimates PM concentrations (average FB= -0.35) whilst the model with FINNpeat

[Figure]

**Figure 4: Daily observed and modelled (a) PM$_{2.5}$ in Singapore, and (b) PM$_{10}$ in Pekanbaru, for WRF-chem runs with different fire emissions inventories and the surface injection option. a) shows observations of PM$_{2.5}$ (solid) and PM$_1$ (dashed). The Pearson's correlation (r) for (a) is 0.47, 0.73, 0.52 and 0.50, and for (b) is 0.63, 0.60, 0.65 and 0.73 for FINN, FINN+GFEDpeat, FINNpeat and FINNpeatSM respectively.**

emissions overestimates PM concentrations (average FB=0.2). The model with FINNpeatSM emissions has the smallest bias

(average FB=-0.11, suggesting mean emissions from this inventory are closest to reality.

[revised manuscript text omitted]

Singapore was from peat fires, slightly higher than the contribution of peat fires to the simulated PM$_{2.5}$ concentration shown

15 in Figure 8 (67%). (Engling et al., (2014) found that in 2006, 76% of particulate matter in Singapore was from peat fires. At the Palangkaraya AERONET site in Kalimantan, (Eck et al., (2019) found that 80-85% of AOD came from peat burning, consistent with the contribution of peat fires to PM$_{2.5}$ simulated by the model at that location (Figure 8)

Inclusion of emissions from peat fires gives largest fire emissions in Sumatra and western Kalimantan, where previously

20 emissions were substantially lower (Fig. 2). This leads to higher PM$_{2.5}$ concentrations across Singapore (Fig. 5), which has a large impact on the population exposure to the PM$_{2.5}$. The location of fires can be an important factor of their contribution to air pollution in populated areas. Lee et al. (2017) found that the 2015 fires in Sumatra accounted for 50% of fire-derived PM$_{2.5}$ in Kuala Lumpur and 41% in Singapore, and (Hansen et al., (2018), found that during August to October fires in South Sumatra and Central Kalimantan are the largest contributors to PM$_{2.5}$ in Singapore. Reddington et al. (2014) and Kim et al. (2015)

[revised manuscript text omitted]

Eck, T.F., Holben, B.N., Giles, D.M., Slutsker, I., Sinyuk, A., Schafer, J.S., Smirnov, A., Sorokin, M., Reid, J.S., Sayer, A.M., Hsu, N.C., Shi, Y.R., Levy, R.C., Lyapustin, A., Rahman, M.A., Liew, S.C., Salinas Cortijo, S. V., Li, T., Kalbermatter, D., Keong, K.L., Yuggotomo, M.E., Aditya, F., Mohamad, M., Mahmud, M., Chong, T.K., Lim, H.S., Choon, Y.E., Deranadyan, G., Kusumaningtyas, S.D.A. and Aldrian, E. 2019. AERONET Remotely Sensed Measurements and Retrievals of Biomass Burning Aerosol Optical Properties During the 2015 Indonesian Burning Season. *Journal of Geophysical Research: Atmospheres*. **124**(8), pp.4722–4740.

Emmons, L.K., Walters, S., Hess, P.G., Lamarque, J., Pfister, G.G., Fillmore, D., Granier, C., Guenther, A., Kinnison, D., Laepple, T., Orlando, J.J., Tie, X., Tyndall, G., Wiedinmyer, C., Baughcum, S.L. and Kloster, S. 2010. Description and evaluation of the Model for Ozone and Related chemical Tracers , version 4 ( MOZART-4 ). *Geoscientific Model Development*., pp.43–67.

Engling, G., He, J., Betha, R. and Balasubramanian, R. 2014. Assessing the regional impact of indonesian biomass burning emissions based on organic molecular tracers and chemical mass balance modeling. *Atmospheric Chemistry and Physics*. **14**(15), pp.8043–8054.

Field, R.D., van der Werf, G.R., Fanin, T., Fetzer, E.J., Fuller, R., Jethva, H., Levy, R., Livesey, N.J., Luo, M., Torres, O. and Worden, H.M. 2016. Indonesian fire activity and smoke pollution in 2015 show persistent nonlinear sensitivity to El Niño-induced drought. *Proceedings of the National Academy of Sciences*. [Online]. **113**(33), pp.9204–9209. Available from: http://www.pnas.org/lookup/doi/10.1073/pnas.1524888113.

Freitas, S.R., Longo, K.M., Chatfield, R., Latham, D., Dias, M.A.F.S., Andreae, M.O., Prins, E., Santos, J.C., Gielow, R. and Carvalho, J.A. 2007. Including the sub-grid scale plume rise of vegetation fires in low resolution atmospheric transport models. *Atmospheric Chemistry and Physics*., pp.3385–3398.

Gaveau, D.L.A., Salim, M. a, Hergoualc'H, K., Locatelli, B., Sloan, S., Wooster, M., Marlier, M.E., Molidena, E., Yaen, H., DeFries, R., Verchot, L., Murdiyarso, D., Nasi, R., Holmgren, P. and Sheil, D. 2014. Major atmospheric emissions from peat fires in Southeast Asia during non-drought years: Evidence from the 2013 Sumatran fires. *Scientific Reports*. **4**.

Gaveau, D.L.A., Sloan, S., Molidena, E., Yaen, H., Sheil, D., Abram, N.K., Ancrenaz, M., Nasi, R., Quinones, M., Wielaard, N. and Meijaard, E. 2014. Four decades of forest persistence, clearance and logging on Borneo. *PLoS ONE*. **9**(7), pp.1–11.

Ge, C., Wang, J. and Reid, J.S. 2014. Mesoscale modeling of smoke transport over the Southeast Asian Maritime Continent : coupling of smoke direct radiative effect below and above the low-level clouds. *Atmospheric Chemistry and Physics*., pp.159–174.

Giglio, L., Randerson, J.T. and Van Der Werf, G.R. 2013. Analysis of daily, monthly, and annual burned area using the fourth-generation global fire emissions database (GFED4). *Journal of Geophysical Research: Biogeosciences*. **118**(1), pp.317–328.

Grell, G.A., Peckham, S.E., Schmitz, R., Mckeen, S.A., Frost, G., Skamarock, W.C. and Eder, B. 2005. Fully coupled '' online ''' chemistry within the WRF model. *Atmospheric Environment*. **39**, pp.6957–6975.

Gruber, A., Dorigo, W.A., Crow, W. and Wagner, W. 2017. Triple Collocation-Based Merging of Satellite Soil Moisture Retrievals. *IEEE Transactions on Geoscience and Remote Sensing*. **55**(12), pp.6780–6792.

Guenther, A., Karl, T., Harley, P., Wiedinmyer, C., Palmer, P.I. and Geron, C. 2006. Estimates of global terrestrial isoprene emissions using MEGAN ( Model of Emissions of Gases and Aerosols from Nature ). *Atmospheric Chemistry and Physics*., pp.3181–3210.

Hansen, A.B., Chong, W.M., Kendall, E., Chew, B.N., Gan, C., Hort, M.C., Lee, S.-Y. and Witham, C.S. 2018. Haze in Singapore – Source Attribution of Biomass Burning from Southeast Asia. *Atmospheric Chemistry and Physics Discussions*., pp.1–29.

Hansen, M.C.C., Potapov, P. V, Moore, R., Hancher, M., Turubanova, S.A. a, Tyukavina, A., Thau, D., Stehman, S.V. V, Goetz, S.J.J., Loveland, T.R.R., Kommareddy, A., Egorov, A., Chini, L., Justice, C.O.O. and Townshend, J.R.G.R.G. 2013. High-Resolution Global Maps of 21st-Century Forest Cover Change. *Science*. [Online]. **342**(November), pp.850–854. Available from: http://www.ncbi.nlm.nih.gov/pubmed/24233722.

Hatch, L.E., Luo, W., Pankow, J.F., Yokelson, R.J., Stockwell, C.E. and Barsanti, K.C. 2015. Identification and quantification of gaseous organic compounds emitted from biomass burning using two-dimensional gas chromatography-time-of-flight mass spectrometry. *Atmospheric Chemistry and Physics*. **15**(4), pp.1865–1899.

Heil, A., Langman, B. and Aldrian, E. 2007. Indonesian peat and vegetation fire emissions : Study on factors influencing large-scale smoke haze pollution using a regional atmospheric chemistry model Indonesian peat and vegetation fire emissions : Study on factors influencing large-scale smoke haze. *Mitigation and Adaption Stratagies for Global Change*. (January).

Hodzic, A. and Jimenez, J.L. 2011. Modeling anthropogenically controlled secondary organic aerosols in a megacity: A simplified framework for global and climate models. *Geoscientific Model Development*. **4**(4), pp.901–917.

[revised manuscript text omitted]

Whitburn, S., Damme, M. Van, Clarisse, L., Turquety, S., Clerbaux, C. and Coheur, P.-. 2016. Peat fires doubled annual ammonia emissions in Indonesia during the 2015 El Niño. *Geophysical Research Letters*.

Wiedinmyer, C., Akagi, S.K., Yokelson, R.J., Emmons, L.K., Al-Saadi, J. a., Orlando, J.J. and Soja, a. J. 2011. The Fire

INventory from NCAR (FINN) – a high resolution global model to estimate the emissions from open burning. *Geoscientific Model Development Discussions*. **3**(4), pp.2439–2476.

Wiggins, E.B., Czimczik, C.I., Santos, G.M., Chen, Y., Xu, X., Holden, S.R., Randerson, J.T., Harvey, C.F., Ming, F. and Yu, L.E. 2018. Smoke radiocarbon measurements from Indonesian fires provide evidence for burning of millennia- aged peat. *Proceedings of the National Academy of Sciences*. **115**(49), pp.12419–12424.

Wooster, M., Gaveau, D., Salim, M., Zhang, T., Xu, W., Green, D., Huijnen, V., Murdiyarso, D., Gunawan, D., Borchard, N., Schirrmann, M., Main, B. and Sepriando, A. 2018. New Tropical Peatland Gas and Particulate Emissions Factors Indicate 2015 Indonesian Fires Released Far More Particulate Matter (but Less Methane) than Current Inventories Imply. *Remote Sensing*. [Online]. **10**(4), p.495. Available from: http://www.mdpi.com/2072-4292/10/4/495.

Wösten, J.H.M., Clymans, E., Page, S.E., Rieley, J.O. and Limin, S.H. 2008. Peat – water interrelationships in a tropical peatland ecosystem in Southeast Asia. . **73**, pp.212–224.

WRI n.d. World Resources Institute. 'Peat lands'. Accessed through Global Forest Watch on [17/04/2017]. Available from: www.globalforestwatch.org.

Zaveri, R.A., Easter, R.C., Fast, J.D. and Peters, L.K. 2008. Model for Simulating Aerosol Interactions and Chemistry ( MOSAIC ). . **113**(July), pp.1–29.